# Integrative differential expression and gene set enrichment analysis using summary statistics for scRNA-seq studies

Ying Ma [1,7], Shiquan Sun [1,7], Xuequn Shang[2], Evan T. Keller [3], Mengjie Chen [4,5] & Xiang Zhou [1,6 ✉]

Differential expression (DE) analysis and gene set enrichment (GSE) analysis are commonly applied in single cell RNA sequencing (scRNA-seq) studies. Here, we develop an integrative and scalable computational method, iDEA, to perform joint DE and GSE analysis through a hierarchical Bayesian framework. By integrating DE and GSE analyses, iDEA can improve the power and consistency of DE analysis and the accuracy of GSE analysis. Importantly, iDEA uses only DE summary statistics as input, enabling effective data modeling through complementing and pairing with various existing DE methods. We illustrate the benefits of iDEA with extensive simulations. We also apply iDEA to analyze three scRNA-seq data sets, where iDEA achieves up to five-fold power gain over existing GSE methods and up to 64% power gain over existing DE methods. The power gain brought by iDEA allows us to identify many pathways that would not be identified by existing approaches in these data.

[1] Department of Biostatistics, University of Michigan, Ann Arbor, MI 48109, USA. [2] School of Computer Science, Northwestern Polytechnical University, Xi'an, Shaanxi 710072, P.R. China. [3] Department of Urology, University of Michigan, Ann Arbor, MI 48109, USA. [4] Department of Human Genetics, University of Chicago, Chicago, IL 60637, USA. [5] Section of Genetic Medicine, Department of Medicine, University of Chicago, Chicago, IL 60637, USA. [6] Center for Statistical Genetics, University of Michigan, Ann Arbor, MI 48109, USA. [7] These authors contributed equally: Ying Ma, Shiquan Sun. ✉email: xzhousph@umich.edu

Single-cell RNA sequencing (scRNA-seq) is becoming a standard technique for transcriptome profiling and is widely applied to many areas of genomics. Compared to the previous bulk RNAseq technique that measures the average gene expression of a potentially heterogeneous cell population, scRNA-seq is capable of producing gene expression measurements both at the genome-wide scale and at a single-cell resolution[1]. Because of the technical advantages, various scRNA-seq studies are being performed to reveal complex cellular heterogeneity in tissues, yielding important insights into many biological processes. However, due to the low amount of mRNAs available in a single cell and the low capture efficiency in the sequencing technique, scRNA-seq data are often extremely noisy[2]. Effective analysis of noisy scRNA-seq data requires development of powerful statistical tools. Here, we develop such a tool for two of the most commonly applied analysis in scRNA-seq studies: differential expression (DE) analysis and gene set enrichment (GSE) analysis.

DE analysis is a routine association analysis task in scRNA-seq studies for identifying genes that are differentially expressed between cell subpopulations, between experimental conditions, or between case control status. Commonly applied DE methods in scRNA-seq include MAST[3], SCDE[4], and zingeR[5], to name a few. While different DE methods make various modelling assumptions to capture diverse aspects of scRNA-seq data[6], almost all of them analyze one gene at a time. Analyzing one gene at a time can lead to potential power loss, as this approach fails to exploit consistent DE evidence across similar genes that could otherwise be used to enhance DE analysis power. It is plausible that due to low statistical power, different scRNA-seq DE methods would tend to prioritize a different set of DE genes in real data applications, leading to sub-optimal performance and inconsistency of results among different methods. In many other types of association analysis such as genome-wide association studies, it has been well recognized that Bayesian approaches that model multiple predictor variables together, even with the simple composite likelihood strategy where information is borrowed across multiple predictor variables each treated independently, can substantially increase power over univariate approaches[7].

GSE analysis is also a routine task that aims to aggregate gene-level DE evidence to the gene set or pathway level. By aggregating gene-level DE evidence, GSE analysis can facilitate the robust biological interpretation of DE results. Many different GSE analysis approaches have been developed, but almost all of them are developed in the bulk RNAseq analysis setting[8]. These existing GSE approaches include over-representation analysis methods, such as DAVID[9] and Fisher's exact test[10]; self-contained test methods, such as $t$-test[11], Chi-square test[12], and others; and competitive test methods such as PAGE[13], GSEA[14], and CAMERA[15]. Despite the abundance of the existing GSE methods, their effectiveness for scRNA-seq analysis remains elusive. Indeed, no comparison studies have been performed thus far to evaluate the effectiveness of the existing GSE methods in the scRNA-seq setting. In addition, and perhaps more importantly, almost all existing GSE methods treat GSE analysis as a separate analytic step after DE analysis. However, GSE analysis and DE analysis are interconnected with each other statistically: while DE results are certainly indispensable for performing GSE analysis to detect enriched gene sets, detected enriched or unenriched gene sets also contain invaluable information that can serve as feedback into DE analysis to enhance its statistical power. Therefore, integrating DE analysis and GSE analysis has the potential to substantially increase the power of both and ensure result reproducibility for scRNA-seq analysis.

Here, we develop a statistical method, which we refer to as the integrative Differential expression and gene set Enrichment Analysis (iDEA), that addresses the aforementioned shortcomings of previous methods for scRNA-seq data analysis. iDEA models all genes together by borrowing information across genes in terms of DE effect size distributional properties. iDEA also integrates DE analysis and GSE analysis into a joint statistical framework, providing substantial power gains for both analytic tasks. Importantly, iDEA makes use of summary statistics output from existing DE tools and does not make explicit modeling assumptions on the individual-level scRNA-seq data. Use of summary statistics not only allows iDEA to take advantage of various existing DE models for effective and flexible data modeling, but also ensures its scalable computation to large-scale scRNA-seq data sets. In addition, incorporating summary statistics from scRNA-seq DE analysis into GSE analysis under the framework of iDEA makes GSE analysis less susceptible to gene-gene correlations and other technical difficulties such as dropout events. We illustrate the benefits of iDEA with extensive simulations and applications to three scRNA-seq data.

## Results

**Methods overview and simulation design.** An overview of iDEA is described in Methods, with technical details provided in Supplementary Notes 1 and 2. We also display a schematic for iDEA in Fig. 1. Briefly, iDEA requires gene-level summary statistics in terms of fold change/effect size estimates and their standard errors as inputs. The input summary statistics can be obtained using any existing scRNA-seq DE methods. As we will show below, given the input from any DE method, iDEA can often improve its power. Besides DE summary statistics, iDEA also requires pre-compiled gene sets. For human data, we have compiled and pruned a total of 12,033 gene sets from seven existing gene set/pathway databases including GO[16], KEGG[17], Reactome[18], BioCarta[19], PubChem Compound[20], ImmuneSigDB[21], and PID[22]. For mouse data, we have compiled and pruned a total of 2851 gene sets from GO[16]. With these inputs, iDEA examines one gene set at a time, performs inference through an expectation maximization algorithm, and uses Louis method[23] to compute a calibrated $p$-value testing whether the gene set is enriched in DE genes or not. In addition, given any gene set, iDEA produces for each gene a posterior probability of DE as its DE evidence. iDEA is implemented as an open source R package, freely available at www.xzlab.org/software.html.

We performed simulations to evaluate the effectiveness of iDEA for GSE analysis and DE analysis (details in Methods). Briefly, we simulated zero-inflated count data for 10,000 genes on 174 cells through a zero-truncated negative binomial distribution using parameters inferred from a real scRNA-seq data. The simulated data shared similar characteristics with the real scRNA-seq data (Supplementary Fig. 1). Among the simulated genes, a certain percentage of them belong to a gene set and we refer to this percentage as the gene set coverage rate (CR). In the null simulations of GSE analysis, each of the 10,000 genes is randomly assigned to be a DE gene with a probability $\exp(\tau_0)/(1 + \exp(\tau_0))$, where $\tau_0$ determines the baseline probability of a gene being DE. Note that the null simulations of GSE analysis contain DE genes, though these DE genes are not enriched in any gene set. In the alternative simulations of GSE analysis, the $j$-th gene is randomly assigned to be a DE gene with probability $\exp(\tau_0 + a_j\tau_1)/(1 + \exp(\tau_0 + a_j\tau_1))$, where $a_j$ is a binary indicator on whether the $j$-th gene belongs to the gene set and $\tau_1$ is the gene set enrichment coefficient that determines whether belonging to the gene set is predictive for the gene being DE. We performed our main simulations in a baseline scenario with $\tau_0 = -2.0$ and CR = 10% and explored different combinations of $\tau_0$, $\tau_1$ and CR to create various simulation scenarios.

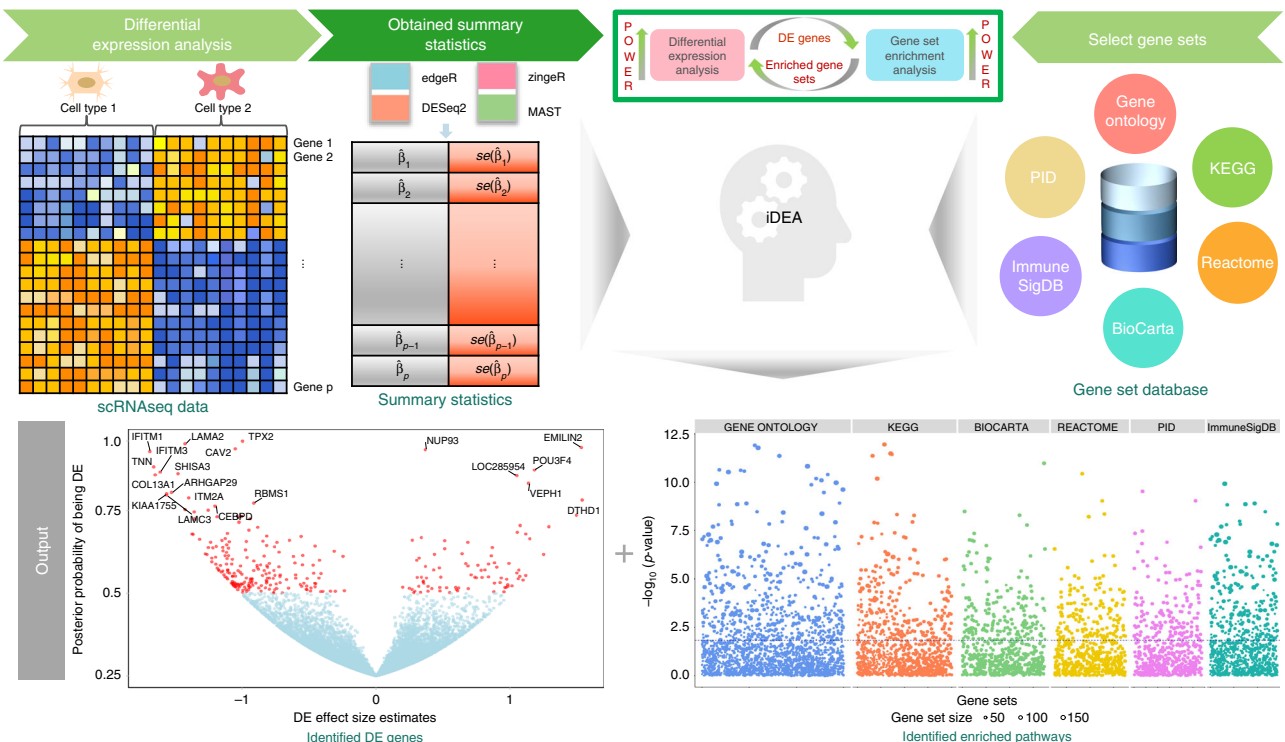

**Fig. 1 Schematic overview of iDEA.** iDEA is designed to jointly model all genes together for integrative differential expression (DE) analysis and gene set enrichment (GSE) analysis. iDEA requires input association summary statistics from existing scRNA-seq DE methods in terms of the DE effect size estimate $\hat{\beta}_j$ and its standard error $se(\hat{\beta}_j)$ for every gene ($j = 1, 2, \cdots, p$) (top left panels). iDEA also requires a pre-defined set of gene sets that we have compiled and pruned for use with the software (top right panels). With these two inputs, iDEA performs joint DE and GSE analysis through a Bayesian hierarchical model. For each gene set, iDEA outputs a $p$-value for testing whether the gene set is enriched with DE genes (bottom right panel) for GSE analysis. In addition, iDEA outputs the posterior inclusion probability of each gene being DE (bottom left panel) for DE analysis. By modeling all genes together and integrating DE and GSE analyses in a joint framework, iDEA can increase the power of both analyses.

**Simulation results.** For GSE analysis, we compared the performance of iDEA with the commonly used GSE analysis methods fGSEA[24], CAMERA[15], PAGE[13], and GSEA[14]. We found that iDEA produces well-calibrated $p$-values under the null in different simulation scenarios (Fig. 2 and Supplementary Fig. 2). The genomic control factor ($\lambda_{gc}$), defined as the ratio between the median empirically observed test statistic and the expected median under the null, is close to one for iDEA across a range of scenarios. Among the other methods, fGSEA, PAGE, and GSEA generally produce calibrated $p$-values (Supplementary Fig. 2); although occasionally the $p$-values from PAGE may slightly deviate from the diagonal line (e.g. when CR = 10% and $\tau_0 = -2.0$; Fig. 2d). In contrast, the $p$-values from CAMERA are only calibrated when CR is very low (1%) and become increasingly overly conservative with increasingly large CR regardless of the DE gene percentage (e.g. Fig. 2b–d; the last two columns in Supplementary Fig. 2). The deflation of CAMERA $p$-values under large CR is presumably because the asymptotic normal approximation used in CAMERA is no longer accurate there. Certainly, we note that under settings with both extremely low $\tau_0$ (e.g. $\tau_0 = -3$; which corresponds to an average of 4.7% genes being DE) and extremely low CR (e.g. CR = 1%), the distribution of $p$-values from all methods would start to deviate from the expected null (e.g. Supplementary Fig. 2, $\tau_0 = -3$, CR = 1%; and to a lesser extent, CR = 5%). Under these extreme parameter combinations, the suboptimal performance of iDEA in terms of type I error control is presumably due to the potential parameter identifiability issue encountered when fitting rare and imbalanced event data[25]. The suboptimal performance of the other methods is presumably because the asymptotic normal

approximation for obtaining $p$-values in these methods becomes no longer accurate.

Besides type I error control, we found that iDEA is more powerful than the other GSE methods across a range of alternative scenarios (Fig. 3a, c and Supplementary Fig. 3). Because different methods have different type I error control, to allow for fair comparison, we computed power at a fixed false discovery rate (FDR) of 5%. In the baseline parameter setting of $\tau_1 = 0.5$ and CR = 10%, we found that iDEA achieved a power of 98%. In contrast, fGSEA, CAMERA, PAGE and GSEA achieved a power of 26%, 0%, 8%, and 26%, respectively (Fig. 3a). The power of iDEA and the other methods all increase with increasing $\tau_1$ as well as with increasing CR (Supplementary Fig. 3). In addition to the power versus FDR plots, the receiver operating characteristic (ROC) curves, displaying false positive rates (FPR) across a range of true positive rates (TPR), also show that iDEA achieves a higher area under the curve (AUC) for GSE analysis (Supplementary Fig. 4). The superior performance of iDEA over existing GSE methods presumably is due to the previously known fact that methods using Kolmogorov Smirnov test (e.g. fGSEA, GSEA) are often not powerful in detecting differences between the distribution of DE test statistics in the gene set versus that outside the gene set, while methods using $t$-tests on the DE z-scores (e.g. CAMERA, PAGE) would also fail to detect gene set enrichment as there is no difference in the mean of DE test statistics in the gene set versus that outside the gene set[26].

For DE analysis, we found that iDEA can improve DE analysis power regardless of whether the summary statistics are from MAST[3], edgeR[5,27] or zingeR[5,28] (Fig. 3b, d, Supplementary Figs. 5, 6). For example, with $\tau_1 = 5$ and CR = 10%, iDEA

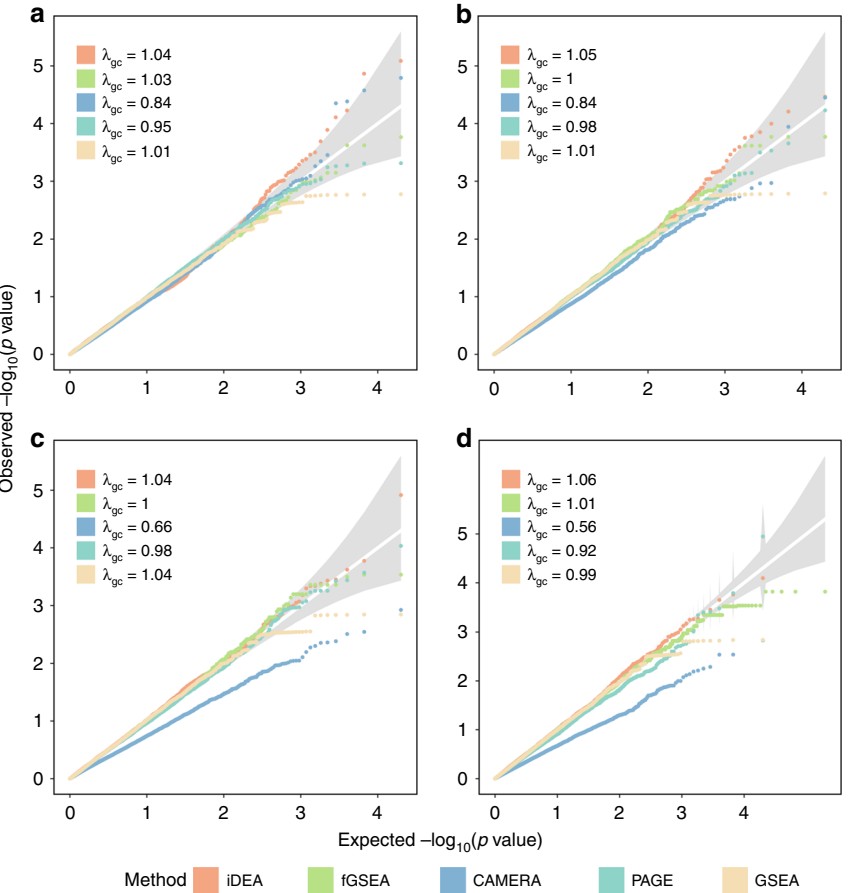

**Fig. 2 iDEA produces well-calibrated *p*-values for gene set enrichment analysis under null simulations.** Quantile-quantile plots of -log10(*p*-values) from iDEA (orange), fGSEA (green), CAMERA (navyblue), PAGE (skyblue) and GSEA (yellow) are shown under different null scenarios with varying gene set coverage rates (CR): CR = 1% (**a**); CR = 2% (**b**); CR = 5% (**c**); CR = 10% (**d**). CR represents the percentage of genes inside the gene set. Here, the other parameters are set to be $\tau_0 = -2$ and $\tau_1 = 0$. $\lambda_{gc}$ is the genomic control factor.

achieves a power of 81%, 61%, and 83% at a true FDR of 5%, when it uses the input summary statistics obtained from zingeR, MAST and edgeR, respectively. In contrast, the power of these three different DE methods are 65%, 52%, and 67%, respectively (Supplementary Figs. 5, 6). The power improvement brought by iDEA is higher in zingeR and MAST than that in edgeR, presumably because the *p*-values from both zingeR and MAST follow approximately a uniform distribution under the null, more so than that from edgeR (Supplementary Fig. 7). Because the model assumption of iDEA also requires the input *p*-values from the DE methods to be well behaved, we will mostly report results based on using zingeR as input in the main text. In the analysis, we also found that the power gain brought by iDEA is mostly due to its joint modeling of DE and GSE analyses, rather than its joint modeling across all genes. Indeed, when the gene set enrichment parameter $\tau_1$ is small, then the power gain brought by iDEA becomes small or negligible (Supplementary Figs. 5, 6). Importantly, iDEA produces reasonably calibrated (or slightly conservative) FDR estimates across a range of simulation scenarios (Supplementary Fig. 8). The ROC curves also yielded consistent results, with iDEA achieving a higher AUC for DE analysis (Supplementary Fig. 4). Besides direct examination of DE analysis power, we also used the Jaccard index to examine the results consistency of different DE methods. Presumably because of the power gain brought up by iDEA, we found that iDEA can also improve the consistency of DE results in terms of the top DE gene list obtained from different methods (Supplementary Fig. 9). For example, when $\tau_1 = 5$ and CR = 10%, the Jaccard index for

the top 1500 genes with the strongest DE evidence obtained by each of three DE methods (MAST, edgeR and zingeR) is 0.59. After applying iDEA to the corresponding summary statistics, the Jaccard index increases to 0.77.

**Human embryonic stem cell scRNA-seq data**. We applied iDEA to analyze three publicly available scRNA-seq data sets. The first scRNA-seq dataset[29] consists of gene expression measurements for 15,280 genes on five cell types (details in Methods). We carried out both GSE and DE analyses on all ten pairs of the five cell types. Because results are largely consistent across different cell type pairs, we mainly report our analysis here on comparing two cell types, DECs and ECs, while list the other comparison results in Supplementary Figs. 10–12.

We first applied iDEA and other GSE methods to detect significantly enriched gene sets across our compiled database of 11,474 human gene sets (Fig. 4a). We also constructed an empirical null *p*-value distribution by permuting the gene labels for each gene set 10 times. Consistent with simulations, we found that the *p*-values in the permuted data from iDEA ($\lambda_{gc} = 1.13$), fGSEA ($\lambda_{gc} = 1.02$), PAGE ($\lambda_{gc} = 0.96$), and GSEA ($\lambda_{gc} = 1.01$) are well behaved, while that from CAMERA show severe deflation ($\lambda_{gc} = 0.29$) (Fig. 4b). For each method, we relied on the empirical null distribution of *p*-values to compute power in detecting enriched gene sets based on a fixed empirical FDR. Consistent with simulations, iDEA identified more significantly enriched gene sets compared to the other GSE methods (Fig. 4c). For example, at an empirical FDR of 5%, iDEA identified 2,106 significantly enriched

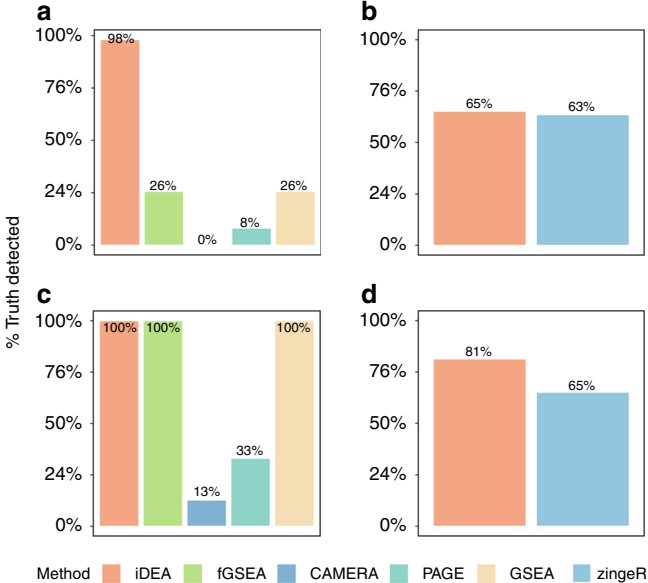

**Fig. 3 iDEA is more powerful for both GSE and DE analyses than existing approaches in power simulations.** The power of iDEA in identifying enriched pathways (y-axis; **a**, **c**) and in identifying differentially expressed genes (y-axis; **b**, **d**) are higher than that of the other methods (x-axis). The compared GSE methods (**a**, **c**) include iDEA (orange), fGSEA (green), CAMERA (navyblue), PAGE (skyblue), and GSEA (yellow). The compared DE methods (**b**, **d**) include iDEA (orange) and zingeR (blue). Simulations are performed under two parameter settings: $\tau_0 = -2$, $\tau_1 = 0.5$, and CR = 10% (**a**, **b**); $\tau_0 = -2$, $\tau_1 = 5$, and CR = 10% (**c**, **d**). Here, power was calculated based on an FDR of 5%.

gene sets, which is 20.9% higher than the next best GSE method (fGSEA, 1,742 significant gene sets). In contrast, CAMERA, PAGE and GSEA identified 537, 1,328, and 1,079 gene sets, respectively. Besides these GSE methods, iDEA is also more powerful than the hypergeometric test (Supplementary Fig. 13). Notably, besides the statistical power, many of the top gene sets identified by iDEA are closely related to embryonic development (Fig. 4e). Examples include the *Wnt* signaling pathway, the transforming growth factor beta (TGF-beta) receptor signaling pathway[30], and relevant GO terms such as GO:0048514 (blood vessel morphogenesis), GO:0001944 (vasculature development) and GO:0007492 (endoderm development)[31]. In order to quantify the biological significance of gene sets identified by different GSE methods, we quantified the relevance between gene sets and embryonic cell development in an unbiased way by searching the related literatures in PubMed (details in Methods). Indeed, in the top 50 enriched gene sets identified by different methods, iDEA identified more gene sets relevant to embryonic cell development (25; Supplementary Table 1) than fGSEA (20), CAMERA (23), PAGE (10), and GSEA (12). The higher number of detected enriched gene sets relevant to embryonic cell development by iDEA provides convergent support for the higher power of iDEA for GSE analysis.

We next applied iDEA for DE analysis where we treated the biologically meaningful gene set GO:0001944 (vasculature development) as the annotation. Consistent with simulations, iDEA identified more DE genes than zingeR. For example, at an empirical FDR of 1%, iDEA identified 2753 DE genes, which is 64.0% higher than zingeR (which identified 1673; Fig. 4d). The 50 selected important DE genes identified by iDEA clearly distinguishes the two examined cell types, DECs and ECs (Fig. 4f). Importantly, based on[29], iDEA identified 1119 genes directly related to definitive endoderm cell differentiation, a process one would expect to be detected by comparing DECs versus ECs; while zingeR only

identified 706. The higher number of DE genes relevant to definitive endoderm cell differentiation detected by iDEA provides convergent support for its higher power for DE analysis. Important DE genes involved in the cell differentiation process that are detected by iDEA but missed by zingeR include *SMAD3*[32], *GATA3*[33], *TGFBR1*[34], *WNT7B*[35], *HAND1*[36], *CCND1*[37], and *HEY2*[38]. Among them, *SMAD3* is essential for activating the necessary transcriptional network for directing definitive endoderm (DE) formation;[32] *GATA3* is indispensable for the signaling pathways in large vessel endothelial cells;[33] *TGFBR1* plays an important role in activating *SMAD2* and *SMAD3*;[34] *WNT7B* is necessary for the redundant ligand–receptor systems which helps activating activate β-catenin signaling in vascular endothelial cells during endoderm development[35]. Finally, as in simulations, iDEA improves the consistency of DE analysis results: the Jaccard index for the top DE genes obtained by each of the three DE methods (MAST, edgeR or zingeR) at an FDR of 1% is only 0.10; after applying iDEA to the corresponding summary statistics, the Jaccard index increases substantially to 0.25 (Supplementary Figs. 14, 15).

**Mouse sensory neuron scRNA-seq data.** The second scRNA-seq data set[39] consists of 13,598 genes and 622 mouse neuronal cells from eleven different cell types. Following the original paper[39], we carried our analysis on comparing the nonpeptidergic nociceptor type I (NP1) neurons with each of the other 10 cell-types (details in Methods). Because the results are again largely consistent across different cell type pairs, we mainly report our analysis here on comparing NP1 versus the remaining ten cell types together. The corresponding results comparing NP1 versus each of the ten cell types are listed in Supplementary Figs. 16–18.

We first applied iDEA for GSE analysis on a pre-compiled set of 2851 mouse gene sets (Fig. 5a). Consistent with simulations, the GSE p-values in the permuted data from iDEA ($\lambda_{gc} = 1.07$), fGSEA ($\lambda_{gc} = 1.08$), PAGE ($\lambda_{gc} = 0.99$), and GSEA ($\lambda_{gc} = 0.94$) are all well behaved, while the p-values from CAMERA show severe deflation ($\lambda_{gc} = 0.11$) (Fig. 5b). Also consistent with simulations, iDEA identified more significantly enriched gene sets compared to the other methods (Fig. 5c). For example, at an FDR of 5%, iDEA identified 1,268 enriched gene sets, which is five times higher than the second-best method (GSEA, 246). In contrast, fGSEA, CAMERA and PAGE identified 236, 134, and 205 enriched gene sets, respectively. Besides these GSE methods, iDEA is also more powerful than the hypergeometric test (Supplementary Fig. 19). Notably, the significant gene sets identified by iDEA are biologically relevant to the compared cell type pair (Fig. 5e and Supplementary Table 2). Most of the top 1% enriched terms were associated with the nervous system, neuronal response and neuronal functions. Such examples include neuron projection (GO:0043005), neuron part (GO:0097458), and somatodendritic compartment (GO:0036477)[40]. Other identified gene sets such as axon (GO:0030424) synapse (GO:0045202) and ion transport (GO:0006811)[41] also play important roles in neuronal functions and activities. None of these gene sets were detected by fGSEA and CAMERA. PAGE and GSEA can also detect these gene sets but do not rank them highly: the rank of these gene sets ranges from top 5% to 61% by PAGE and from top 15% to 71% by GSEA. In addition, use of iDEA recovered 102 out of the 237 gene sets known to be involved in inflammatory itch[39]. In contrast, fGSEA, CAMERA, PAGE, and fGSEA identified 31, 20, 19, and 29 gene sets among them, respectively.

We next applied iDEA for DE analysis where we treated the gene set GO:0097458 (neuron part) as the annotation. Again, iDEA identified more DE genes than zingeR (Fig. 5d). At an FDR of 1%, iDEA detected 1,103 DE genes, which is 11.0% higher than zingeR (993). We illustrate 50 selected DE genes identified by iDEA in

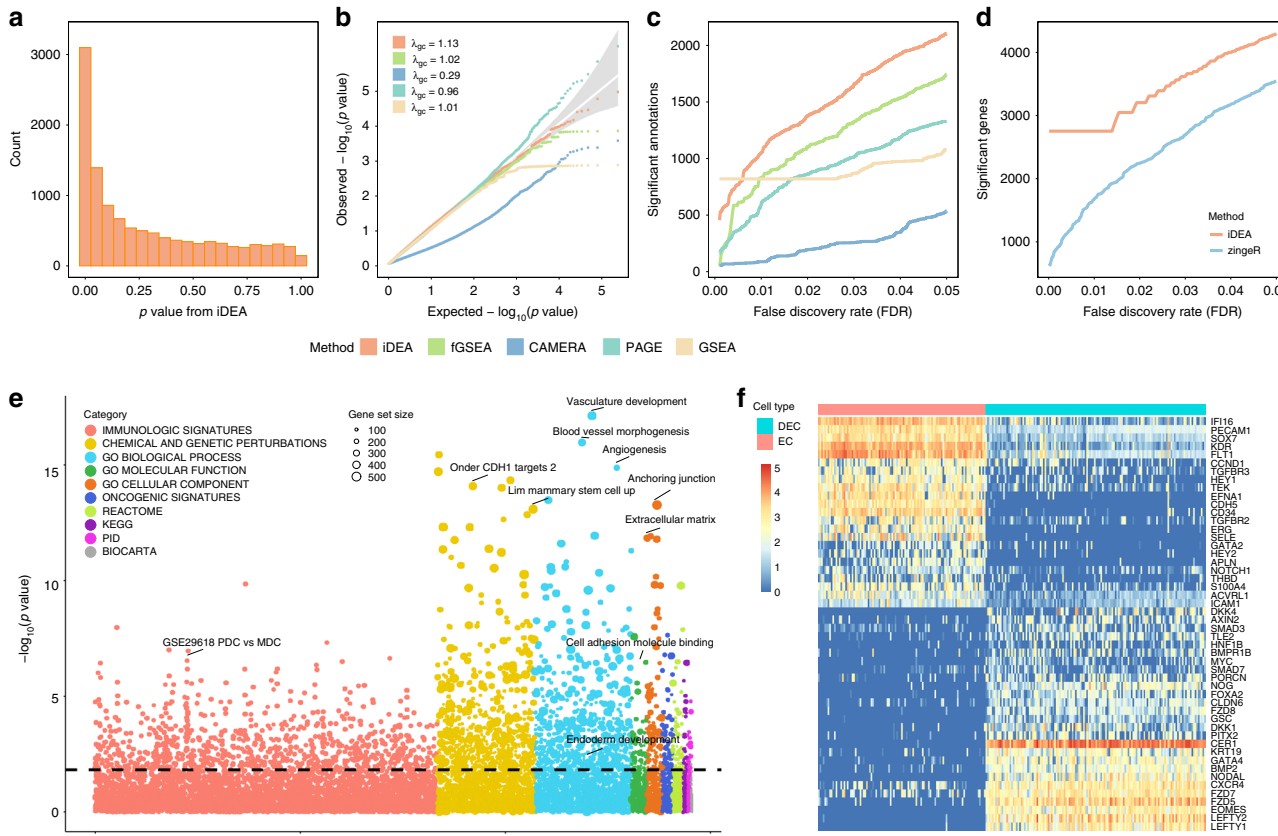

**Fig. 4 Analysis results in the embryonic stem cell scRNA-seq data.** Results are shown for comparing two cell types, endothelial cell (EC) and definitive endoderm derivatives cell (DEC). **a** $p$-values from iDEA for GSE analysis display expected enrichment of small $p$-values (for true signals) and a long flat tail toward large $p$-values. **b** Quantile-quantile plots of $-\log10(p\text{-values})$ from GSE methods including iDEA (orange), fGSEA (green), CAMERA (navyblue), PAGE (skyblue) and GSEA (yellow) are shown under permuted null. The $p$-values from iDEA, fGSEA, PAGE and GSEA are reasonably well calibrated, while that from CAEMRA is overly conservative. Here $\lambda_{gc}$ is the genomic control factor. **c** Number of identified enriched gene sets by iDEA (orange), fGSEA (green), CAMERA (navyblue), PAGE (skyblue) and GSEA (yellow) are plotted against different empirical false discovery rates (FDR). iDEA is more powerful than other methods for GSE analysis. **d** Number of identified DE genes by iDEA (orange) and zingeR (blue) are plotted against different empirical FDR values. iDEA is more powerful than zingeR for DE analysis. **e** Heatmap shows the normalized expression level (log10-transformation with pseudo-count 0.1) for selected 50 DE genes (rows) identified by iDEA for cells in the two cell types (columns). Genes are sorted by Hierarchical clustering, cells are ordered by cell types (EC: red; DEC: blue). These DE genes clearly distinguish two compared cell types. **f** Bubble plot shows $-\log10(p\text{-values})$ for GSE analysis from iDEA ($y$-axis) for different gene sets. Gene sets are colored by ten categories: immunologic signatures (red), chemical and genetic perturbations (yellow), GO biological process (blue), GO molecular function (green), GO cellular component (orange), oncogenic signatures (deep blue), Reactome (grass-green), KEGG (purple), PID (rose), and Biocarta (gray). The size of the dot represents the number of genes contained in the gene set. Names for ten of the gene sets that are closely related to embryonic cell development are highlighted in the panel.

Fig. 5f, which clearly distinguish the two compared cell types. Many NP1 neuron marker genes are identified by iDEA but missed by zingeR even at an FDR of 5%. These marker gene examples include *MRGPRB5*, *STX1B*, *FAM167A*, *KLK8*, and *STK32A*. Among these genes, *MRGPRB5* is a Mas-related gene expressed in primary nociceptive sensory neurons[42]. *KLK8* mediates signals in the PAR1-dependent signaling responses in the nociceptive neurons[43]. Importantly, iDEA detected 79 DE genes out of top 100 previously known NP1 DE genes listed in the original study, while zingeR detected 75 DE genes, again supporting the high power of iDEA. Finally, consistent with simulations, iDEA also improves the consistency of DE analysis results; namely, the Jaccard index for the top DE genes obtained by each of the three DE methods (MAST, edgeR or zingeR) at an FDR of 1% is 0.14; after applying iDEA to the corresponding summary statistics, the Jaccard index increases to 0.17 (Supplementary Figs. 20, 21).

**10x Genomics PBMC scRNA-seq data.** The third scRNA-seq data set consists of 13,713 genes and 2638 cells collected from

peripheral blood mononuclear cells (PBMCs)[44]. We focused on comparing CD4+ T-cells with CD8+ T-cells in order to examine the performance of various methods in the challenging setting where the two examined cell types are similar (Supplementary Fig. 22). We also focused on examining a small set of 144 gene sets that contain important gene signatures of immune and stroma cell types[45]. In particular, these gene sets contain CD4+ and CD8+ cell type signatures and thus can be treated as true positives for method comparison in this data.

We first applied iDEA to identify enriched gene signatures among these true positives (Fig. 6a). Due to the small number of gene sets examined here, the $p$-values from all methods in the permuted data are not discernable from the null expectation (Fig. 6b). Likely due to the low read depth in 10x genomics data and the subsequent high gene expression measurement noise, all GSE methods have similar power in terms of detecting enriched gene sets based on a fixed FDR threshold (Fig. 6c). However, almost all top enriched gene sets identified by iDEA are relevant to CD4 or CD8 cell functions (Fig. 6e). For example, in the top 25

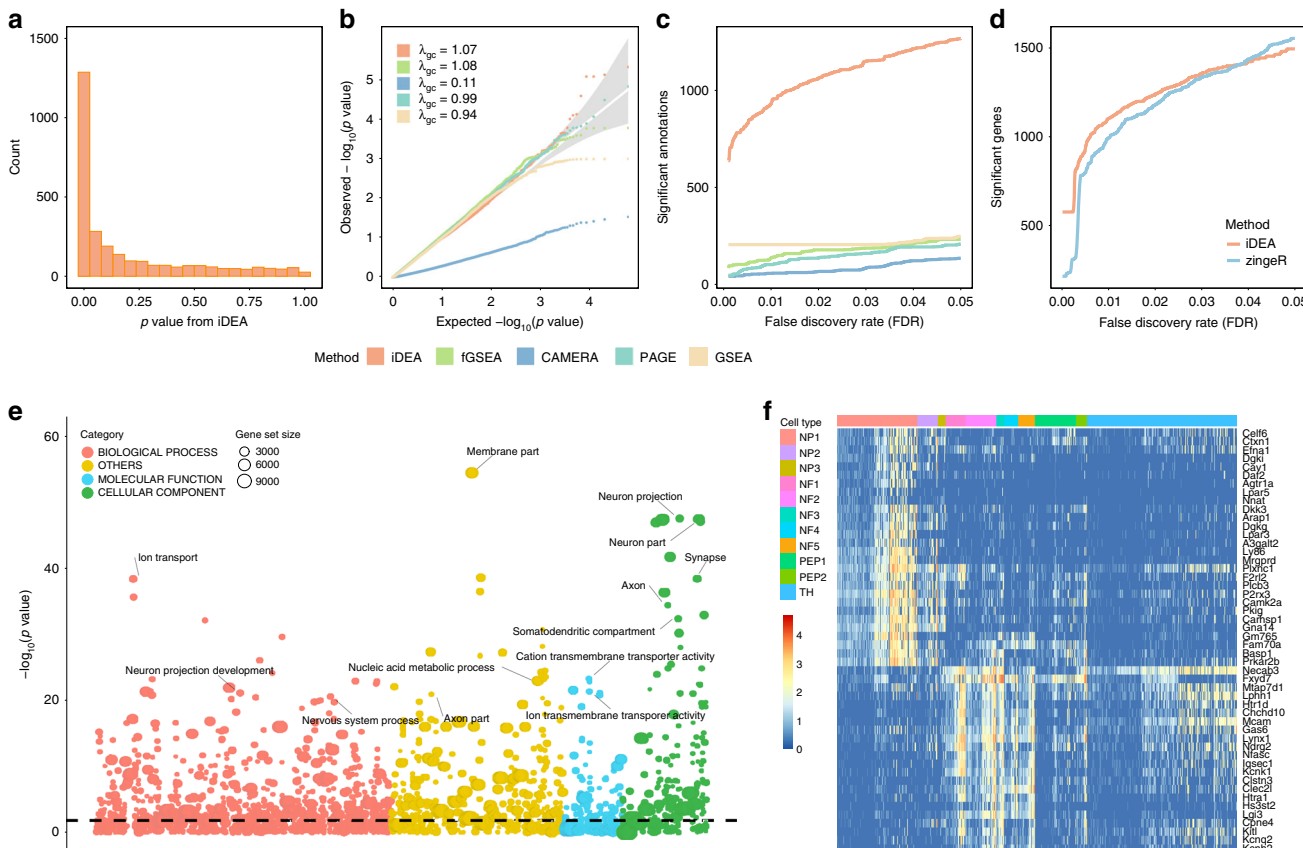

**Fig. 5 Analysis results in the mouse neuronal cell scRNA-seq data.** Results are shown for comparing nonpeptidergic nociceptors 1 (NP1) versus all the other cell types. **a** p-values from iDEA for GSE analysis display expected enrichment of small p-values (for true signals) and a long flat tail towards large p-values. **b** Quantile-quantile plots of $-\log10$(p-values) from GSE methods including iDEA (orange), fGSEA (green), CAMERA (navyblue), PAGE (skyblue) and GSEA (yellow) are shown under permuted null. The p-values from iDEA, fGSEA, PAGE and GSEA are reasonably well calibrated, while that from CAEMRA are overly conservative. Here $\lambda_{gc}$ is the genomic control factor. **c** Number of identified enriched gene sets by iDEA (orange), fGSEA (green), CAMERA (navyblue), PAGE (skyblue) and GSEA (yellow) are plotted against different empirical false discovery rates (FDR). iDEA is more powerful than other methods for GSE analysis. **d** Number of identified DE genes by iDEA (orange) and zingeR (blue) are plotted against different empirical FDR values. iDEA is more powerful than zingeR for DE analysis. **e** Heatmap shows the normalized expression level (log10-transformation with pseudo-count 0.1) for selected 50 DE genes (rows) identified by iDEA for cells in the two cell types (columns). Genes are sorted by Hierarchical clustering, cells are ordered by cell types (NP1: red; others: other colors). These DE genes clearly distinguish two compared cell types. **f** Bubble plot shows $-\log10$(p-values) for GSE analysis from iDEA (y-axis) for different gene sets. Gene sets are colored by four categories: GO biological process (orange), GO molecular function (blue), GO cellular component (green) and other gene ontology terms with only GO numbers (yellow). The size of the dot represents the number of genes contained in the gene set. Names for ten of the gene sets that are closely related to nociceptive sensory neurons' activities are highlighted in the panel.

gene sets identified by iDEA, 22 of them are relevant to CD4 or CD8 cells (Supplementary Table 3). In contrast, 13 from fGSEA (Supplementary Table 4), 13 from CAMERA (Supplementary Table 5), 14 from PAGE (Supplementary Table 6), and 13 from GSEA (Supplementary Table 7) are relevant to CD4 or CD8 cells. Besides these commonly used GSE methods, iDEA is also more powerful than the hypergeometric test, which only identified one significant gene set (Supplementary Fig. 23).

We next applied iDEA to perform DE analysis where we treated the gene set CD8 + T-effector memory as the annotation. Consistent with simulations, iDEA identified more DE genes than zingeR (Fig. 6d). At an FDR of 1%, iDEA detected 255 significant DE genes, which is 15.3% higher than that detected by zingeR (221). We illustrate 30 selected DE genes identified by iDEA (Fig. 6f), which clearly distinguish the two cell types. The significant DE genes identified by iDEA include *CD8A, KLRG1, GNLY,* and *PRF1* that are all relevant to CD T cell differentiation[46]. Indeed, iDEA identified many T cell activation and

differentiation related genes that are missed by zingeR. Among the genes missed by zingeR, *BTG2* is important for T-cell activation marker expression, T cell proliferation and migration[47]; *KLF2* is involved in both the activation of CD4+ T cell trafficking (through regulation of *S1PR1*) and T helper cells differentiation;[48] *CD247* of the Ctex region is essential for the TCR-mediated activation of T cells;[49] and *LSP1* is found to be down regulated in human T-cell lines and plays an important role in the process of T-cell transformation[50]. iDEA also improved the consistency of DE analysis results. Specifically, the Jaccard index for the top DE genes obtained by each of three DE methods (MAST, edgeR or zingeR) at an FDR of 1% was only 0.06; after applying iDEA to the corresponding summary statistics, the Jaccard index increased substantially to 0.15 (Supplementary Fig. 24, 25).

Finally, while the DE analysis relies on a pre-selected gene set, we found that the number of DE genes identified by iDEA without the pre-selected gene set (=252, at an FDR of 1%) is

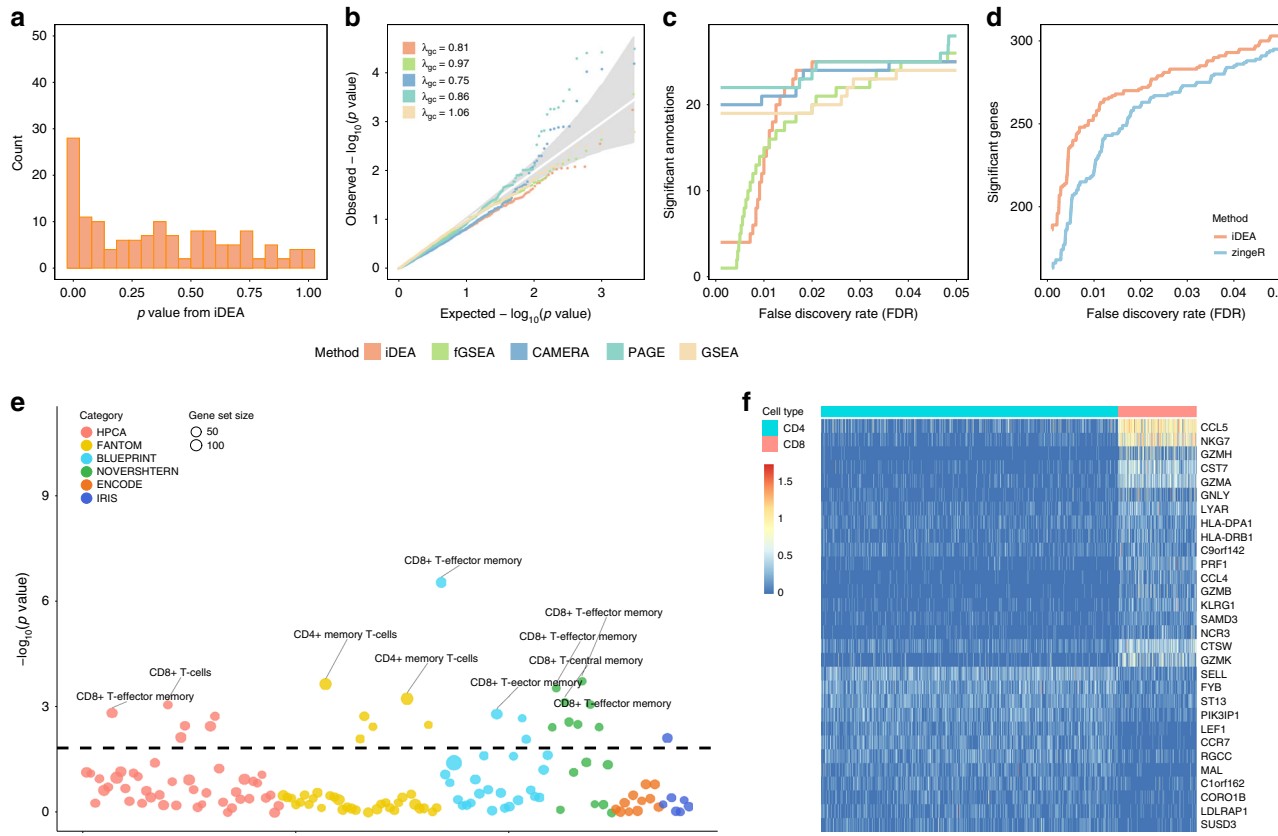

**Fig. 6 Analysis results in the 10X Genomics scRNA-seq data.** Results are shown for comparing CD4+ T cells versus CD8+ T cells. **a** *p*-values from iDEA for GSE analysis display expected enrichment of small *p*-values (for true signals) and a long flat tail towards large *p*-values. **b** Quantile-quantile plots of $-\log_{10}$(*p*-values) from GSE methods including iDEA (orange), fGSEA (green), CAMERA (navyblue), PAGE (skyblue) and GSEA (yellow) are shown under permuted null. The *p*-values from all methods in the permuted data are not discernable from the null expectation. Here $\lambda_{gc}$ is the genomic control factor. **c** Number of identified enriched gene sets by iDEA (orange), fGSEA (green), CAMERA (navyblue), PAGE (skyblue) and GSEA (yellow) are plotted against different empirical false discovery rates (FDR). iDEA is as the same powerful than other methods for GSE analysis. **d** Number of identified DE genes by iDEA (orange) and zingeR (blue) are plotted against different empirical FDR values. iDEA is more powerful than zingeR for DE analysis. **e** Heatmap shows the normalized expression level (log10-transformation with pseudo-count 0.1) for selected 30 DE genes (rows) identified by iDEA for cells in the two cell types (columns). Genes are sorted by Hierarchical clustering, cells are ordered by cell types (CD4: blue; CD8: red). These DE genes clearly distinguish two compared cell types. **f** Bubble plot shows –log10(*p*-values) for GSE analysis from iDEA (y-axis) for different gene sets. Gene sets are colored by six projects: HPCA (red), FANTOM (yellow), BLUEPRINT (blue), NOVERSHTERN (green), ENCODE (orange), IRIS (deep blue). The size of the dot represents the number of genes contained in the gene set. Names for ten of the gene sets that are closely related to CD4+ and CD8+ immune process are highlighted in the panel.

similar to the results using the cell type defined gene set (=255), both are larger than that identified by zingeR (=221) (Supplementary Fig. 26). Using the large set of human gene sets described earlier, instead of using the small cell type specific gene sets, also leads to a similar conclusion on the type I error control and power of iDEA (Supplementary Fig. 27). Indeed, many top enriched gene sets identified by iDEA from the large gene sets are also related to immune systems (Supplementary Table 8). In the three real data applications, we also found that the results from iDEA are largely insensitive to the choice of hyperparameters in the prior distribution for the variance parameters (analysis details in Methods; Supplementary Fig. 28). Performing analysis on two groups of cells randomly selected form the same cell type also demonstrates the proper type I error control by iDEA (Supplementary Fig. 29).

## Discussion

We have presented a computational method, iDEA, for integrating DE analysis and GSE analysis in scRNA-seq studies. iDEA directly models summary statistics from existing scRNA-seq DE tools, produces well-calibrated *p*-values for enriched gene set detection, and provides increased power for both DE and GSE analyses. Modeling summary statistics in iDEA circumvents the need for explicit modeling of individual-level scRNA-seq data, allowing iDEA to be paired with existing DE tools for quick adaptation across a range of scRNA-seq data types. In addition, use of summary statistics makes iDEA reasonably computationally efficient, with its computing complexity scaling linearly with respect to the number of genes (Supplementary Table 9). We have demonstrated the benefits of iDEA using both simulations and applications to three recently published scRNA-seq data sets.

We have primarily focused on scRNA-seq data, as we aimed to perform a comprehensive comparative study on GSE methods for scRNA-seq studies in addition to developing iDEA. However, the flexible modeling framework of iDEA can be equally applied to bulk RNA-seq studies. To illustrate this, we applied iDEA to an oral carcinoma bulk RNA-seq dataset[51], where we show that iDEA can identify more DE genes and more enriched gene sets that are relevant to oral carcinogenesis (Supplementary Fig. 30; Supplementary Note 3).

We have primarily focused on modeling the marginal effect size estimates and standard errors from DE analysis, which is equivalent to modeling of marginal z-scores. Our modeling of z-scores follows that of[52,53] and effectively assumes that the prior distribution of true effect sizes is dependent on the standard errors, and subsequently the sample size (Supplementary Note 4). Such prior dependence on sample size appears to have relatively mild consequences in practical data analysis and has attractive theoretical properties[54]. Nevertheless, we have developed a variant of iDEA that does not require prior dependence on sample size. The iDEA variant has similar performance as the original iDEA in the real data applications, properly controlling for type I error and displaying higher power than the other methods (Supplementary Figs. 31–34; Supplementary Note 4).

The DE analyses in our real data applications are performed by treating a pre-selected gene set as annotation based on prior biological knowledge. Certainly, selecting such gene set may not always be possible in every study. In the absence of a pre-selected gene set to serve as the annotation, we developed a Bayesian model averaging (BMA) approach (Supplementary Note 5) to aggregate DE evidence across all available gene sets. The BMA approach yields consistent results for majority of genes as compared to the pre-selection approach in the real data applications (Supplementary Fig. 35), demonstrating its utility for practical applications.

iDEA does not explicitly account for gene set overlap that may cause non-independence among gene sets. In practice, we found that the gene set overlap is generally small: the median number of overlapped genes among pairs of gene sets in the human data is only 1 (5 in the mouse data), as compared to the median gene set size of 143 (131 in the mouse data). A careful examination of the top identified enriched gene sets in the real data applications also suggest that gene set overlap does not appear to introduce excessive false signals (Supplementary Tables 10, 11; Supplementary Note 6). In addition, the sparse data structure in scRNA-seq appears to further diminish the concern on gene-gene correlations. Indeed, GSE methods that do not explicitly account for gene-gene correlation (e.g. iDEA, PAGE, fGSEA and GSEA) appear to provide more calibrated p-values than methods that explicitly account for gene-gene correlation (e.g. CAMERA) in these real data applications. Nevertheless, we followed most existing GSE approaches and accounted for GSE test non-independence due to gene set overlap through permutation of gene labels. Such permutation retains the gene set overlap proportion under the empirical null: if one gene set contains genes that are overlapped with genes in another gene set in the real data, then the overlapped number remains the same in the permuted data. Consequently, the test statistics on the two gene sets would be correlated in a somewhat comparable fashion between the permuted data and the real data. By estimating FDR based on such permuted null, we can account for test non-independence due to gene set overlaps.

Finally, we acknowledge that general caveat exists for DE analysis between cell types in scRNA-seq studies: because cell types are often inferred based on the whole gene expression matrix, DE analysis performed on the inferred cell types may lead to inflated DE test statistics with artificially smaller standard errors[55]. We have attempted to alleviate such issue by conducting our analysis on datasets where cell types are reasonably rigorously defined and validated through other experiments (Supplementary Note 7). Nevertheless, future methodological innovations are needed to account for the uncertainty associated with cell type inference for DE analysis between cell types in scRNA-seq studies.

## Methods

**iDEA overview**. Here, we provide a brief overview of iDEA, with technical details provided in Supplementary Notes 1, 2. iDEA models all genes jointly and requires summary statistics from standard DE analysis for all genes. These summary statistics are in the form of marginal DE effect size estimate $\hat{\beta}_j$ and its standard error $\text{se}\left(\hat{\beta}_j\right), j = 1, 2, \cdots p$, where $p$ is the number of genes. We assume that the estimated DE effect size centers around the true effect size $\hat{\beta}_j \sim N\left(\beta_j, \text{se}\left(\hat{\beta}_j\right)^2\right)$, and that the true effect size $\beta_j$ follows a mixture of two distributions depending on whether $j$-th gene is a DE gene or not:

$$\beta_j \sim \pi_j N\left(0, \text{se}\left(\hat{\beta}_j\right)^2 \sigma_\beta^2\right) + \left(1 - \pi_j\right)\delta_0, \tag{1}$$

where $\pi_j$ is the prior probability of being a DE gene; $\sigma_\beta^2$ is a scaling factor that determines the DE effect size strength; and $\delta_0$ is the Dirac function that represents a point mass at zero. Therefore, with proportion $\pi_j$, $j$-th gene is a DE gene and its DE effect size $\beta_j$ follows a normal distribution with a large variance $\text{se}\left(\hat{\beta}_j\right)^2 \sigma_\beta^2$. With proportion $1 - \pi_j$, $j$-th gene is a non-DE gene and its DE effect size is exactly zero. Note that our modeling above is also equivalent to modeling using marginal z-statistics

$$z_j \sim \pi_j N\left(0, \sigma_\beta^2 + 1\right) + \left(1 - \pi_j\right)N(0, 1), \tag{2}$$

where $z_j$ is the marginal z-score on the DE evidence for the $j$-th gene.

In Eq. (1), we have scaled the variance with respect to $\text{se}\left(\hat{\beta}_j\right)^2$ using the scaling factor $\sigma_\beta^2$, so that our analysis results are scale invariant; that is, the results remain the same regardless what the DE effect size is measured on.

For the scaling factor, we followed existing statistical literature and chose the conjugate distribution for a variance parameter as the prior for $\sigma_\beta^2$. Specifically, we specify an inverse gamma prior on $\sigma_\beta^2$: $\sigma_\beta^2 \sim \text{InvG}(a_\beta, b_\beta)$ with $a_\beta = 3.0$, $b_\beta = 20.0$, which ensures a prior mean of 10 ($= b_\beta/(a_\beta-1)$) and the existence of a prior variance (which requires $a_\beta > 2$). To integrate the gene set information into the above model, we model the gene-specific probability of being a DE gene as

$$\text{logit}\left(\pi_j\right) = \log\left(\frac{\pi_j}{1 - \pi_j}\right) = \tau_0 + a_j \tau_1 \tag{3}$$

where $\tau_0$ is an intercept that determines the proportion of DE genes outside the gene set; $a_j$ is a binary indicator on whether $j$-th gene belongs to the gene set ($a_j = 1$) or not ($a_j = 0$); and $\tau_1$ is a gene set enrichment parameter that determines the odds ratio of DE for genes inside the gene set versus genes outside the gene set. To facilitate computation, we introduce a vector of binary indicators $\gamma = \left(\gamma_1, \cdots, \gamma_p\right)^T$ to indicate whether each gene is a DE gene ($\gamma_j = 1$) or not ($\gamma_j = 0$). Therefore, the prior distribution of $\gamma_j$ is effectively a Bernoulli distribution,

$$\gamma_j \sim \text{Bern}\left(\pi_j\right). \tag{4}$$

With proportion $\pi_j$, $j$-th gene is a DE gene and with proportion $1 - \pi_j$, $j$-th gene is a non-DE gene and its DE effect size is exactly zero. With the above model setup, we are primarily interested in inferring two parameters: the gene-specific indicator $\gamma_p$, which indicates whether $j$-th gene is a DE gene or not; and the enrichment parameter $\tau_1$, which represents the enrichment of DE genes in the gene set. We aim to infer the posterior probability of $\gamma_j = 1$ as evidence for $j$-th gene being DE and test the null hypothesis $H_0 : \tau_1 = 0$ that DE genes are not enriched in the gene set.

To achieve both goals, we develop an expectation maximization (EM)-Markov chain Monte Carlo (MCMC) algorithm for parameter estimation. Briefly, we treat the vector of both $\beta$ and $\gamma$ as missing data and develop an iterative EM optimization algorithm that alternates between an expectation step and a maximization step. In the expectation step, the expectation of the log likelihood effectively requires computing the posterior probability of each gene being a DE gene, $P(\gamma_j = 1|\text{data})$, through MCMC. In the maximization step, we estimate the enrichment parameter through optimization, which is effectively equivalent to fitting a logistic regression model, where we treat the posterior probabilities for each gene being DE obtained from the expectation step as the outcome variable. While our EM-MCMC algorithm yields accurate parameter estimates, we found that the standard errors for the enrichment parameter obtained through the complete-data log-likelihood function is overly liberal and leads to p-value inflations ($\lambda_{\text{gc}} = 1.33$; Supplementary Fig. 36A), a phenomenon that has been observed in many other settings[23,56]. Therefore, we used the Louis method[23] to obtain the corrected information matrix and produce calibrated p-values ($\lambda_{\text{gc}} = 1.06$; Supplementary Fig. 36B).

**Summary statistics and gene annotations**. iDEA requires DE summary statistics in the form of fold change/effect size estimates and their standard errors as input. These summary statistics in principal can be obtained using any existing scRNA-seq DE methods, such as MAST[3] or zingeR[5] etc. Here, we primarily focus on

presenting the results obtained based on input from zingeR, which directly outputs DE effect size estimates and their standard errors and which is the most recent DE method for scRNA-seq analysis. However, we also explored the benefits of pairing iDEA with different scRNA-seq DE methods in part of the Results section. Details of these DE methods are provided in the next subsection.

In addition to DE summary statistics, iDEA also requires pre-defined gene sets. For human data, we downloaded a total of 12,033 gene sets based on seven existing gene set/pathway databases annotated on the reference genome GRCh37 from MSigDB databases (http://software.broadinstitute.org/gsea/downloads.jsp). These databases include BioCarta[19], KEGG[17], GO[16], PubChem Compound[20], ImmuneSigDB[21], PID[22], and Reactome[18]. We divided the compiled gene sets into ten functional categories that include immunologic signatures (4856 gene sets), chemical and genetic perturbations (2,379 gene sets), GO biological process (2835 gene sets), GO molecular function (544 gene sets), GO cellular component (355 gene sets), oncogenic signatures (186 gene sets), Reactome (415 gene sets), KEGG (163 gene sets), PID (207 gene sets), and Biocarta (93 gene sets). We merged the gene sets with summary statistics and filtered out gene sets that contain less than 20 genes and finally focused on a total of 11,474 gene sets in GSE analysis. For mouse data, we downloaded the gene ontology (GO) annotations of mouse genes in the GAF 2.0 format from the website (http://www.informatics.jax.org/downloads/reports/index.html#go). We merged the gene sets with summary statistics and filtered out gene sets that contain less than 50 genes and finally focused on a total of 2851 gene sets in GSE analysis. These GO terms were based on four categories: biological process (1719 gene sets), cellular component (279 gene sets), molecular function (297 gene sets), and unannotated gene sets (556 gene sets). For 10x Genomics data, we collected a total of 489 gene sets consisting of cell type specific gene signatures from Xcell and these gene signatures were previously collected to annotate 64 distinct cell types and cell subsets[45]. We merged the gene sets with summary statistics and filtered out gene sets that contain less than 10 genes to focus on a final set of 144 cell type specific gene sets. We note that we filtered out gene sets with a small number of genes (e.g.<10 or <20) as the pruning step due to computational reasons: for iDEA and other GSE methods, the gene set enrichment parameter estimation can become inaccurate and unstable for gene sets with small sizes.

**Compared methods**. For DE analysis in both simulations and real data applications, we compared iDEA with three existing approaches: (1) MAST (version 1.8.1), which outputs a coefficient (coef) as the effect size estimate and a corresponding $p$-value for each gene[3]; (2) edgeR (version 3.8), which outputs a log fold change (logFC) as the effect size estimate and a corresponding $p$-value for each gene[27]. The edgeR function we used was the weighted version of edgeR: it first calculated the cell-level weights using ZINB-WaVE and then used these weights inside edgeR for final computation; (3) zingeR (version 1.0)[5], where we applied zingeR to obtain cell-level weights, which were further supplied to DESeq2 (version 1.18.1) for DE analysis. The output from this procedure consists of a log fold change (log2FoldChange) as the effect size estimate and a corresponding standard derivation (lfcSE) for every gene[28]. iDEA can be paired with each of these DE methods to use the corresponding summary statistics as input for analysis. In these DE methods, in order to extract summary statistics for iDEA, we treated either the logarithm fold change or fold change as the gene effect size $\hat{\beta}_j$, and back derived the standard error of $\hat{\beta}_j$ using the unsigned z-score as se$\left(\hat{\beta}_j\right) = |\hat{\beta}_j/\text{zscore}|$, where z-score was either directly available or was obtained by transforming the $p$-value via the R function qnorm($p$-value/2.0, lower.tail = F). Afterwards, we used summary statistics obtained from these DE methods to fit iDEA.

For GSE analysis in both simulations and real data applications, we mainly compared iDEA with four existing approaches: (1) fGSEA (R version 1.8.0)[24]; (2) CAMERA (inside limma, R version 3.8.3)[15]; (3) PAGE (PGSEA, R version 1.56.0)[13]; and (4) GSEA (Java version 2.2.4)[14]. We used z-score statistics from zingeR DE analysis as input for all these methods. Here, the z-score statistics were calculated by the transformation of the unadjusted $p$-values, paired with the sign of log-fold change estimate: z score $= \Phi^{-1}\left(1 - \frac{p\,\text{value}}{2}\right)\text{sign}(\text{logFC})$, where $\Phi(\cdot)$ denotes the standard Gaussian cumulative distribution. We used the default settings for all GSE methods. We used the recommended interGeneCorrelation function in CAMERA to calculate the correlation between genes. In addition, we compared iDEA with the hypergeometric test in all real data applications. In particular, we counted the number of DE genes (defined as $p$-value < 0.05) and non-DE genes in the gene set as well as outside the gene set and performed hypergeometric test to obtain GSE $p$-value.

Note that, in the GSE analysis, we have primarily focused on comparing our method with traditional GSE methods that aim to identify gene sets whose genes are differentially expressed between cell types or treatment conditions. Different from these traditional GSE methods, several methods have been recently developed for scRNA-seq studies that are targeted for a completely different enrichment task: identifying gene sets whose genes show coordinated transcriptional heterogeneity. Exemplary such methods include the pathway and gene set overdispersion analysis (PAGODA)[57] and f-scLVM[58]. Because both PAGADA and f-scLVM are targeted for detecting coordinated expression heterogeneity (i.e. gene-gene correlation within a gene set) rather than the usual GSE analysis based on DE analysis, we did not compare our method and other GSE methods with them.

**Simulations**. We performed simulations to evaluate the performance of iDEA for both DE analysis and GSE analysis. In each simulation replicate, we simulated 10,000 genes. We randomly assigned a proportion of these genes to belong to a gene set of interest. We referred to the percentage of genes belonging to the gene set as the coverage rate (CR), which were set to be either 1%, 2%, 5%, or 10%; where 10% is close to the median CR of all analyzed pathways in the present study. We further introduced a binary indicator $a_j$ to represent whether $j$-th gene belongs to the gene set ($a_j = 1$) or not ($a_j = 0$). Afterwards, we randomly assigned each gene to be a DE gene with probability $\pi_j$, which depends on $a_j$. In particular, the parameter $\pi_j$ is in the form of

$$\pi_j = \frac{\exp\left(\tau_0 + a_j\tau_1\right)}{1 + \exp\left(\tau_0 + a_j\tau_1\right)}, \tag{5}$$

where the intercept parameters $\tau_0$ was set to be either $-0.5$, $-1.0$, $-2.0$, or $-3.0$ to present different proportions of DE genes in the data (e.g. $\tau_0 = -2$ represents that roughly 12% of genes are DE genes; $\frac{\exp(-2)}{1+\exp(-2)} \approx 0.12$); while the gene set enrichment coefficient $\tau_1$ was set to be either 0 (no enrichment of DE genes in the gene set), 0.25 (weak enrichment), 0.5, 1.0 (moderate enrichment), or 5.0 (strong enrichment). Note that the median gene set enrichment parameter estimate across all analyzed pathways in the real data applications is close to 0.5 while the highest enrichment parameter estimate is 17.

In order to compare the performance of iDEA of DE analysis with other count-based DE methods, we simulated scRNA-seq gene expression counts first. To make our simulations as realistic as possible, the simulations were performed based on parameters inferred from a published scRNA-seq data[29]. Specifically, to simulate scRNA-seq gene expression counts, we selected two cell types that include endothelial cells (EC; 105 cells) and trophoblast-like cells (TB; 69 cells) from Chu et al.[29]. We fitted each gene using a zero-truncated negative binomial (ZTNB). Through the ZTNB model, we first inferred the gene-specific mean expression parameter $\lambda_j$ and dispersion parameter $\phi_j$ in the negative binomial component of ZTNB through method of moments[5]. In particular, these parameter estimates are obtained iteratively through

$$\lambda_j^{(t+1)} = \frac{\sum_i Y_{ij}\left(1 - f_{NB}\left(\lambda_j^{(t)} N_i, \phi_j^{(t)}\right)\right)}{\sum_i N_i} \tag{6}$$

$$\phi_j^{(t+1)} = \frac{\sum_i \left(\lambda_j^{(t)} N_i\right)^2}{\sum_i Y_{ij}^2\left(1 - f_{NB}\left(\lambda_j^{(t)} N_i, \phi_j^{(t)}\right)\right) - \sum_i \left(\lambda_j^{(t)} N_i\right)^2 - \sum_i \left(\lambda_j^{(t)} N_i\right)}, \tag{7}$$

where $Y_{ij}$ is the non-zero expression count for $i$-th cell and $j$-th gene in the real data (note that we ignored zero counts in this estimation step); $N_i$ denotes the total read counts (i.e. read depth) for $i$-th cell; the superscripts $(t)$ and $(t+1)$ denote the $t$-th and $(t+1)$-th iteration estimates, respectively; $f_{NB}(\cdot, \cdot)$ is the negative binomial density function.

In addition, we also followed[5] to infer the zero proportion parameters $p_{ij}$ in the ZTNB model by borrowing information across all genes. Specifically, we model the dropout probability for $i$-th cell and $j$-th gene, $p_{ij}$, using a semi-parametric additive logistic regression model:

$$z_{ij} \sim \text{Bern}(p_{ij}), \tag{8}$$

$$\log\frac{p_{ij}}{1 - p_{ij}} = s(A_j) + \log(N_i) + s(A_j)\log(N_i), \tag{9}$$

where $z_{ij}$ is an indicator on whether the observed count for $i$-th cell and $j$-th gene is zero or not; Bern($p_{ij}$) denotes a Bernoulli distribution with the dropout parameter $p_{ij}$; s($\cdot$) is a non-parametric thin-plate spline; $A_j$ is the average logarithm scale counts per million (CPM) calculated by aveLogCPM function in edgeR[27]. This way, the dropout probability becomes both cell-specific and gene-specific.

With the above estimated parameters, we simulated gene count through ZTNB model for both DE and non-DE genes. For DE genes, we simulated each DE effect size from a normal distribution with mean zero and standard deviation 3.5. For non-DE genes, we directly set the DE effect size to be zero. We then calculated the true fold change of each gene as the exponential of effect size fc $= \exp(\beta_j)$, which is multiplied to the estimated mean gene expression levels $\hat{\lambda}_j$ in one population, resulting in a mean of $\hat{\mu}_{ij} = \hat{\lambda}_j \text{fc}_j$ for all cells in one population and a mean of $\hat{\mu}_{ij} = \hat{\lambda}_j$ for all cells in the other population. Afterwards, we simulated count data for $i$-th cell and $j$-th gene, $C_{ij}$, follows a negative binomial distribution NB$\left(\hat{\mu}_{ij}, \hat{\phi}_j\right)$. We set $C_{ij}$ to be exactly zero with probability $\hat{p}_{ij}$. Note that the simulations do not exactly match the iDEA modeling assumptions, allows us to examine the robustness of iDEA.

We simulated the gene expression counts with $\tau_0$ fixed to be $-2$, and with varying $\tau_1$ (i.e., 0, 0.25, 0.5, 1.0, or 5.0) and varying CR (i.e., 1%, 2%, 5%, or 10%). With the simulated count data, we fitted different DE methods to obtain summary statistics, with which we fitted iDEA and other GSE methods. We evaluated the power of DE analysis and GSE analysis. To evaluate the type I error control of

different GSE methods, we examined the null simulation settings ($\tau_1 = 0$). In each null setting, we permuted the gene labels 10 times to construct the permuted null sets, to which we applied different GSE methods. We then calculated the genomic inflation factor ($\lambda_{gc}$) for each GSE methods. Here, the genomic inflation factor ($\lambda_{gc}$) is defined as the ratio of the median of the empirically observed distribution of the test statistic to the expected median. Specifically, we first convert the $p$-value for the gene sets to chi-squared test statistics then calculated $\lambda_{gc}$ by dividing the resulting chi-squared test statistics by the expected median of a chi-squared distribution with one degree of freedom (0.4549364; qchisq(0.5,1) in R). To evaluate the power of different GSE methods, in each simulation setting, we obtained 1,000 simulation replicates with enriched pathways (i.e., $\tau_1 \neq 0$) and 9,000 simulation replicates without enriched pathways (i.e., $\tau_1 = 0$). We then evaluated the power of GSE analysis in detecting these 1000 true signals given an FDR of 5%. To evaluate the power of different DE methods, we again computed power to detect true DE genes based on an FDR of 5%.

**ScRNA-seq data sets.** We applied iDEA to analyze three published scRNA-seq data sets. The first scRNA-seq data is from Chu et al.[29] (GEO accession number GSE75748). It contains a total of 19,097 genes on 1018 cells from seven cell types. The seven cell types include the human embryonic stem (ES) cell with subtypes H1 (212 cells) and H9 (162 cells); four ES derived linear-specific progenitor cell types that include neuronal progenitor cell (NPC, ectoderm derivatives, 173 cells), definitive endoderm derivatives cell (DEC, 138 cells), endothelial cell (EC, mesoderm derivatives, 105 cells), trophoblast-like cell (TB, extraembryonic derivatives, 69 cells); and human foreskin fibroblasts cell (HFF, 159 cells). We focused our analysis on five ES derived cell types (NPCs, DEs, ECs, TBs, and HFFs) and examined all pairs among them. For each pair, we filtered out lowly expressed genes that have more than 5 counts in at most two cells. The resulting number of analyzed genes ranges from 14,918 (EC vs TB) to 15,778 (DEC vs NPC). We considered batch information as a covariate when we fit zingeR to obtain summary statistics. For DE analysis of iDEA, we included the gene set vasculature development which is known to be important for vasculature progression and endothelial cell development[59]. To evaluate GSE analysis results, we examined the top 50 significant gene sets identified by each GSE methods. To obtain the unbiased evaluation of different GSE methods, we used the R package RISmed (version 2.1.7) to query the related articles with the keywords: gene set name, cell type, and "embryonic development". We input one gene set at a time, and the number of gene set that do has the relevant literatures is counted to quantify the performance of different GSE methods.

The second scRNA-seq data is from Usoskin et al.[39] (GEO accession number GSE59739). This dataset contains a total of 19534 genes on 622 neuronal cells collected from the mouse lumbar dorsal root ganglion. These cells were classified into 11 neuronal cell types from four categories. The cell types include the neurofilament containing (NF) category: NF1 (31 cells), NF2 (48 cells), NF3 (12 cells) and NF4 (22 cells), NF5 (26 cells); nonpeptidergic nociceptors (NP) category: NP1 (125 cells), NP2 (32 cells), and NP3 (12 cells); peptidergic nociceptors (PEP) category: PEP1 (64 cells) and PEP2 (17 cells); and tyrosine hydroxylase containing (TH; 233 cells) category. NPs cell type versus the remaining cell types are shown in main results; while the pairs of NP1 cell type with each of the other ten cell types are shown in Supplementary Figs. 16–18. For each pair, we filtered out lowly expressed genes that have more than 5 counts in at most two cells. The resulting number of analyzed genes ranges from 10,009 (comparing NP1 cell type vs NP3 cell type) to 10,948 (comparing NP1 cell type vs NF2 cell type). We included picking sessions as a covariate when we fit zingeR to obtain summary statistics[5]. For DE analysis of iDEA, we used the biological meaningful gene set neuron part (GO:0097458)[40] in the model.

The third scRNA-seq data is a peripheral blood mononuclear cells (PBMCs) data obtained from 10x Genomics website (https://support.10xgenomics.com/single-cell-gene-expression/datasets/1.1.0/pbmc3k)[44]. We downloaded the filtered gene/cell matrix that contains 2,700 cells and 32,738 genes. We processed the data using the R package Seurat[60] following the tutorial (https://satijalab.org/seurat/pbmc3k_tutorial.html) to obtain at a final set of 2,638 cells and 13,713 genes. We obtained clustering results from Seurat as shown in Supplementary Fig. 22. Here, we focus our analysis on 1153 CD4+ T-cells and 305 CD8+ T-cells, to examine the performance of various methods in the challenging setting where the two examined cell types are similar to each other. We obtained summary statistics from zingeR and filtered out genes with $p$-values larger than 0.8 to focus on a final set of 1,696 genes. We did this due to the $p$-values obtained by the zingeR under the null are seriously left-skewed distributed. For DE analysis of iDEA, we used the gene signature CD8+ T-effector memory[61] in the model.

In the real data applications, for both DE analysis and GSE analysis, we calculated power of different methods based on estimated FDR through permutations. Specifically, for DE analysis, we permuted the cell type label across cells ten times. We then applied different DE methods and obtained the empirical null distribution of test statistics ($p$-values or posterior estimates of $\gamma$'s), with which we calculated the empirical FDR for each threshold. For iDEA, in the permuted data, we also fixed the gene set enrichment parameters $\hat{\tau}$ to be those estimated in the real data without re-estimating them. In our experience, re-estimating the enrichment parameters can lead to overly liberal FDR estimates and slows computation. For GSE analysis, we permuted the gene set label across all genes for

each gene set ten times. We then applied different methods and obtained the enrichment $p$-values in the permuted null, with which we further calculated the empirical FDR for each $p$-value threshold.

**Sensitivity analysis.** In the main real data applications, we have fixed the hyperparameters for the inverse Gamma distribution ($a_\beta = 3.0$, $b_\beta = 20.0$, to ensure a prior mean $\frac{b_\beta}{a_\beta - 1}$ of 10) because there is insufficient information to estimate these parameters. Specifically, the inverse Gamma distribution serves as the prior for the variance parameter, which can be estimated by the effect sizes across many DE genes. Because there is only one variance parameter, it is impossible to estimate the hyperparameters in the inverse Gamma distribution for this variance parameter. Therefore, instead of estimating these hyperparameters, we performed sensitivity analysis to examine whether results would change with respect to the hyperparameters. To do so, we varied the hyperparameters and tested across a range of gene sets with different coverages in the three real datasets. Specifically, we varied the hyperparameters so that the prior mean of the inverse gamma distribution is 0.001, 0.1, 1, 10, 100. We also varied the coverage rate to be the 10th, 30th, 50th, 70th, 90th percentile of the gene set size of the gene sets we used for the corresponding real data analysis. For example, for the human embryonic scRNA-seq dataset, we pick the gene set with coverage rate to be the 10th, 30th, 50th, 70th, 90th percentile of the gene set size of the human gene sets we analyzed and set the hyper parameter in the prior distribution of $\sigma_\beta^2$, $(a_\beta, b_\beta)$ to be (3, 0.02), (3, 0.2), (3, 2), (3, 20), (3, 200), respectively. For the mouse sensory neuron scRNA-seq dataset and 10x Genomics PBMC scRNA-seq data, we followed the same procedure as the first scRNA-seq dataset.

**Ethics approval and consent to participate.** No ethical approval was required for this study. All utilized public data sets were generated by other organizations that obtained ethical approval.

**Reporting summary.** Further information on research design is available in the Nature Research Reporting Summary linked to this article.

## Data availability

The datasets used in the present study are all publicly available. The human embryonic stem cell scRNA-seq dataset is available at [https://www.ncbi.nlm.nih.gov/geo/query/acc.cgi?acc=GSE75748]. The Mouse sensory neuron scRNA-seq data is available at [http://linnarssonlab.org/drg/]. The 10x Genomics PBMC scRNA-seq data is available at 10x Genomics website [https://support.10xgenomics.com/single-cell-gene-expression/datasets/1.1.0/pbmc3k]. For the gene sets we collected, the human gene sets are available from MSigDB databases [http://software.broadinstitute.org/gsea/downloads.jsp] and the mouse gene sets are available from the website [http://www.informatics.jax.org/downloads/reports/index.html#go]. In addition, all raw data and processed data used for analysis are also available at [https://github.com/xzhoulab/iDEA].

## Code availability

The iDEA software package and source code have been deposited at [www.xzlab.org/software.html]. All scripts used to reproduce all the analysis is also available at the same website.

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

## Acknowledgements

This study was supported by the National Institutes of Health (NIH) Grants R01HG009124 (to X.Z.) and R01GM126553 (to M.C.), and the National Science Foundation (NSF) Grant DMS1712933 (to X.Z.). S.S. was supported by NIH Grant R01HD088558 (PI Tung). After moving back to China, S.S. is also supported by the National Natural Science Foundation of China (Grant No. 61902319) and the Natural Science Foundation of Shaanxi Province (Grant No. 2019JQ127). E.T.K. is supported by NIH Grants P01CA093900 and P30CA046592 by use of the Cancer Center Single Cell Analysis Resource. We thank Dr. Xingjie Hao at Huazhong University of Science and Technology for helping with the collection of gene sets.

## Author contributions

X.Z. conceived the idea and provided funding support. Y.M., S.S., and X.Z. designed the experiments, with input from X.S., E.K., and M.C. Y.M. and S.S. developed the method,

implemented the software, performed simulations and analyzed real data. Y.M., S.S., and X.Z. wrote the paper with input from all other authors.

## Competing interests

The authors declare no competing interests.
