## [Peer Review File · Nature Communications]

Reviewers' Comments:

Reviewer #1:

Remarks to the Author:

The authors describe a new method for gene set enrichment analysis, that considers both the effect size in form of log₂ fold changes (LFC) as well as the significance of gene expression change i.e. the standard error of the LFC. This is achieved by implementing an EM-MCMC method with the LFCs and their s.e. estimated using any DE-method and a gene-set annotation as input. The method then computes the posterior probability of a gene being DE, given the previous evidence as well as the DE-status of other genes within the same set, while simultaneously optimizing an enrichment parameter for the set. This then results in both a test statistic for the enrichment of DE-genes within a set as well as an updated DE statistic for each gene.

This method should be applicable to both single cell and bulk RNA-seq data (probably also for microarrays). The method is then evaluated using simulations and is also applied to three scRNA-seq data sets. Generally, I like the idea to use both effect size and significance in GSEA and I believe that this method has great potential.

Major points:

It is commendable that the authors also developed simulations to benchmark their method. I don't understand why the authors then only report the TPR comparison and do not consider FPR or FDR. Instead of the barplot in Figure 3 I would prefer to see something like a ROC-curve.

The method is advertised in particular for use with single cell RNA-seq data, however, the method is only compared to older GSEA methods originally developed for bulk or microarray data. I would appreciate some mentioning of more specific single cell methods such as PAGODA or slalom.

Alternatively, the authors could also include a bulk RNA-seq data set and illustrate the potential utility of iDEA to evaluate the effects of GC-content and gene length biases.

The authors mention that they use 'pruned' GO or Reactome data as groups, however, I was not able to find out how the data were pruned and how pruning affects the estimation of the enrichment parameters? This is of particular interest, since the package only provides annotation for GRCh37.

In GO, one gene can be a member of multiple groups, which complicates the testing due to non-independence. How does iDEA deal with this problem? Some examples of tests for nested groups would be interesting to see.

In the same vein, it is unclear how the authors construct gene sets based on 10xGenomics signatures. To my knowledge these signatures are mainly used for the classification of cells and it is unclear to me how these signatures would be used to infer the biological function of genes.

Minor comments:

In several figures where axes are labelled as p-value, but it is $-\log_{10}(\text{p-value})$?

Supplementary Figure 13: Because iDEA increases the numbers the total number of detected genes, I find it difficult to compare the numbers in the Venn Diagrams. Could the authors provide some percentages or use upsetR plots?

Reviewer #2:

Remarks to the Author:

In this manuscript, the authors describe a new computational method for gene set enrichment analysis. The proposed iDEA algorithm uses a hierarchical Bayesian model with EM-MCMC estimation of the parameters to obtain p-values for enrichment of DE genes within each gene set. The authors go on to demonstrate improved performance compared to several other DE and GSEA

methods on real and simulated data. I found the hybrid of frequentist and Bayesian concepts rather bizarre, but that notwithstanding, I found many other causes for concern with the statistical aspects of this work. These issues are described in more detail below.

MAJOR:

1. It seems that the authors are using their proposed method to detect differences between cell types. However, the effect sizes are often overestimated and the standard errors understated for such comparisons, as the "cell types" are not known *a priori* but are instead defined from the expression values. (A more detailed description of this effect is available at [http://bioconductor.org/packages/3.9/workflows/vignettes/simpleSingleCell/inst/doc/de.html#5_caveats_with_interpreting_de_\(p\)-values](http://bioconductor.org/packages/3.9/workflows/vignettes/simpleSingleCell/inst/doc/de.html#5_caveats_with_interpreting_de_(p)-values).) Thus, the authors' focus on error control in simulations is largely irrelevant to practical analyses, as the underlying effect sizes and their standard errors cannot be considered as unbiased estimates.
2. The authors use the standard error of $\hat{\beta}$ in the variance term for their distribution of β in Equation 1. This makes little sense. The assumed distribution of the true effect size should not depend on the properties of its estimate; this is akin to saying that the truth changes depending on how we estimate it, which is generally absurd (quantum physics aside). From a practical perspective, the assumed distribution shrinks as the number of cells increases due to improved precision of the effect size. This seems as if it would reduce the power to identify DE genes, as the likelihood of a large effect size being generated from the normal component goes to zero. One would have instead hoped that more cells would provide greater power to detect DE.
3. The prior distribution for σ_{β} is poorly justified, and I could not find an analysis of the sensitivity of the method to changes in prior parameters. This is important as σ_{β} effectively determines the likelihood by which a gene is considered to be DE, which in turn serves as the basis of the gene set enrichment aspect of the method. In addition, the prior distribution for the indicator variables γ are not described, despite the fact that the authors are computing posterior distributions for $\gamma \in \{0, 1\}$ via MCMC.
4. It is not clear to me how the proposed method considers a gene to be DE if the calculations are always performed in the context of gene sets. Are the authors calling genes to be DE if they are called as DE in any calculation with any gene set? This seems to run into major problems with multiple testing.
5. The authors criticize existing methods as treating DE and GSEA analyses as two separate steps. However, it seems that the same thing is done here! The proposed method relies on the effect size and the standard error, which can be trivially transformed into a per-gene p-value. At this point, the DE analysis has effectively been performed already, so iDEA is really just doing the second-stage gene set enrichment analysis.
6. In the proposed EM-MCMC step, I do not understand how the posterior probabilities can be used as the outcome of a logistic regression. Surely the response must be categorical - is the logistic model's likelihood even valid for continuous values? If it is valid, that leads to the next question - are the standard errors of the posterior probabilities taken into consideration by the subsequent inference steps, or are users expected to run the MCMC for sufficient iterations that the standard errors are assumed to be close to zero?
7. The authors focus heavily on the number of significant gene sets in their analyses of real datasets, which serves as evidence for iDEA's improved power over existing methods. However, as per point 1, this is largely irrelevant. The main focus is instead on the ranking of the top gene sets, as only these are used for biological interpretation. Even if we detect >1000 significant gene sets, no analyst will look at the 1000th gene set if the first ~100 sets are not useful for determining cell type or state.

8. The authors criticize methods like FGSEA and CAMERA on the basis of their inability to detect differences between genes inside and outside the test set. However, it seems that related criticisms would apply to iDEA. For example, iDEA would not detect a gene set where the enrichment of DE genes is the same in terms of the number of genes, but the effect sizes of the genes in the set are systematically higher than those outside of the set. One might say that detecting such differences is not the purpose of iDEA, but in that case, it would be more appropriate to compare iDEA to conceptually similar tests like the hypergeometric test and its derivatives (e.g., goseq).

9. I could not obtain p-values from the software when testing on a small simulated example. Running code adapted from <https://xzhoulab.github.io/iDEA/> just gave me an empty output data.frame in idea@gsea.

```
> library(iDEA)
>
> set.seed(100)
> ngenes <- 10000
> summary_data <- cbind(
> log2FoldChange=rnorm(ngenes),
> lfcSE2=rchisq(ngenes, 1)/100
> )
> rownames(summary_data) <- sprintf("GENE_%i", seq_len(ngenes))
>
> annotation_data <- matrix(0, ngenes, 2)
> annotation_data[head(order(summary_data[,1]), 100),1] <- 1
> annotation_data[sample(ngenes, 100), 2] <- 2
>
> colnames(annotation_data) <- LETTERS[1:ncol(annotation_data)]
> rownames(annotation_data) <- sprintf("GENE_%i", seq_len(ngenes))
>
> idea <- CreateiDEAObject(summary_data, annotation_data, num_core=2)
> idea <- iDEA.fit(idea)
> idea <- iDEA.louis(idea)
```

MINOR:

- The use of the expit() transformation in the simulation design is not explained. Why not use a simpler approach where a given proportion of genes are set as DE?

Reviewer #3:

Remarks to the Author:

Sun et al. propose iDEA, a unified statistical framework that jointly performs differential expression (DE) and gene set enrichment (GSE) analysis. The method is certainly welcome in the field of transcriptomic studies. Analysis is carried out to evaluate performance via both simulations and empirical studies. The paper is overall well written. My comments are below.

1) The summary statistics from DE analysis are used as input for iDEA. I wonder why the authors restrict their analysis/method to the single-cell setting. Of course, the model for scRNA-seq would be much different than that for bulk RNA-seq, as the authors have pointed out. However, the noise modeling etc specific to scRNA-seq is handled by the respective existing methods adopted (e.g., MAST, zingeR). The distinguishing features of iDEA that specific to scRNA-seq need to be clearly stated and emphasized.

2) Across all simulation runs and real data analyses, iDEA returns more significant genes/gene sets. The authors only looked at type I error control via QQ plots of p-values and showed that the p-values by iDEA were well calibrated. (BTW, λ_{gc} , the genomic control factor, needs to be explicitly defined.) What are the actual type I error rates, since this is from simulation run? Also, it has been well acknowledged that model-based simulated sc data do not reflect the true observed sc data. The authors need to demonstrate iDEA controls for type I error by performing test under the true null using two data sets of the same cell types/tissues/model organisms.

3) The two human empirical studies adopted drastically different number of gene sets (12,033 vs 144). Based on this, I have a few comments/concerns.

a. Do the users need to prune the gene sets themselves? Guidelines and discussions on this will be helpful.

b. The 12,033 gene sets were compiled from seven different databases. This number is on the same order of magnitude as the number of genes. Are there overlaps between the gene sets? Will this bias the FDR control procedures, i.e., will the testing across different gene sets be non-independent?

c. For the 10X Genomics dataset, if using all 12,033 gene sets, will it be underpowered?

d. For the 144 gene sets, the authors mentioned "we focused on examining a small set of 144 gene sets that contain important gene signatures of immune and stroma cell types..." Is there a cyclic problem? Usually, the gene sets are defined specific to both the tissue of interest and the study design (what are the treatment etc), and the gene set enrichments can be identified through the DE results. Here, the authors implicitly imposed a "prior" (by focusing on a small number of related gene sets based on, e.g., previous DE results), performed the analysis using this "prior", and then tested the DE based on results. Will the results be simply passed down from the strong "prior"?

4) The GitHub page contains only functions with very limited and sketchy descriptions. For this method to be suited for publication at Nature Communications, a package needs to be compiled with detailed documentation and user manual, preferably with toy datasets.

5) On line 80, the authors discussed the limitations of existing univariate DE methods. It sounded as if iDEA would adopt a multivariate model, which is not true – iDEA still takes as input testing results based on univariate models.

6) Are the hyperparameters for the inverse Gamma fixed? Can they be empirically estimated and thus method/dataset specific?

Response to Reviewer #1

The authors describe a new method for gene set enrichment analysis, that considers both the effect size in form of log₂ fold changes (LFC) as well as the significance of gene expression change i.e. the standard error of the LFC. This is achieved by implementing an EM-MCMC method with the LFCs and their s.e. estimated using any DE-method and a gene-set annotation as input. The method then computes the posterior probability of a gene being DE, given the previous evidence as well as the DE-status of other genes within the same set, while simultaneously optimizing an enrichment parameter for the set. This then results in both a test statistic for the enrichment of DE-genes within a set as well as an updated DE statistic for each gene.

This method should be applicable to both single cell and bulk RNA-seq data (probably also for microarrays). The method is then evaluated using simulations and is also applied to three scRNA-seq data sets. Generally, I like the idea to use both effect size and significance in GSEA and I believe that this method has great potential.

Responses: Thank you for your positive review and constructive comments. Our detailed responses are listed below.

Major points:

1. It is commendable that the authors also developed simulations to benchmark their method. I don't understand why the authors then only report the TPR comparison and do not consider FPR or FDR. Instead of the bar plot in Figure 3 I would prefer to see something like a ROC-curve.

Response: Thank you for the comment. We apologize for a potential miscommunication here. The power/TPR of all simulations (e.g. that in Figure 3 and S3) are indeed calculated based on a fixed false discovery rate (FDR). These details were previously provided in the Simulation Results section (currently lines 221-222 on page 8, lines 247–249 on page 9, lines 855-860 on page 27-28). Following your comments, we have modified the Figures 3, S3, S5 and S6 title to make this information more explicit. Certainly, we previously only focused on plotting power/TPR at relatively small FDR cutoff values (0.05), since these values were most relevant to practical analysis. Following your comments, we further added additional results using ROC-curves. The ROC based results are largely consistent with previous FDR based results: iDEA achieved a higher Area Under the Curve (AUC) in both GSE analysis and DE analysis. For example, in the baseline parameter setting of

$\tau_1 = 0.5$ and CR = 10%, we found that iDEA achieved an AUC of 0.99. In contrast, fGSEA, CAMERA, PAGE and GSEA achieved an AUC of 0.83, 0.62, 0.55 and 0.82, respectively (Figure S4A). In the DE analysis, for example, with $\tau_1 = 5$ and CR = 10%, iDEA achieves an AUC of 0.96 when using summary statistics from zingeR while zingeR achieves 0.9 (Figure S4D). These ROC results are shown in the new Supplementary Figure S4, with details explained in the Results section (lines 234-237 on page 9, lines 265-266 on pages 9-10).

2. The method is advertised in particular for use with single cell RNA-seq data, however, the method is only compared to older GSEA methods originally developed for bulk or microarray data. I would appreciate some mentioning of more specific single cell methods such as PAGODA or slalom. Alternatively, the authors could also include a bulk RNA-seq data set and illustrate the potential utility of iDEA to evaluate the effects of GC-content and gene length biases.

Response: Thank you for your comment. We previously only compared our method with traditional GSE methods because all these methods are focused on standard GSE analysis – identifying gene sets whose genes are differentially expressed between cell types or treatment conditions. In contrast, both the pathway and gene set overdispersion analysis (PAGODA) [1] and f-scLVM [2] (implemented in the package “slalom”) are completely different enrichment methods that aim to identify genes sets whose genes show coordinated transcriptional heterogeneity. Specifically, PAGODA examines one gene set at a time, extracts the first eigen value from the gene expression matrix defined by the gene set, and tests whether the first eigen value is larger than expected under the null where the gene expression matrix defined by the gene set is a random matrix. f-scLVM further jointly analyze multiple gene sets together to identify gene sets that can explain variance in the gene expression matrix. Note that f-scLVM can only be used to analyze tens to hundreds of gene sets/pathways, and both of PAGODA and f-scLVM do not require differential expression analysis. Therefore, both PAGADA and f-scLVM are targeted for detecting coordinated expression heterogeneity (i.e. gene-gene correlation within a gene set) rather than the usual GSE analysis. Because these methods are for a completely different task, we are not able to compare our method and other GSE methods to PAGODA or slalom (note that the PAGODA and slalom papers also did not compare their method with the traditional GSE methods). Instead, following your suggestion, we have added description on these two important methods and explanation on why we did not compare to them in the Materials and Methods section (lines 767-778 on page 25).

In addition, following your suggestion, we have also included a bulk RNA-seq dataset for illustration. We illustrate how iDEA can be useful for analyzing bulk data, with a side

analysis on the effects of GC-content and gene length biases. New results are shown in Supplementary Figures S30 and are briefly mentioned in the Discussion section (lines 521-526 on page 17) with details provided in the Supplementary Text (section 3, from page 41 to page 43).

3. The authors mention that they use ‘pruned’ GO or Reactome data as groups, however, I was not able to find out how the data were pruned and how pruning affects the estimation of the enrichment parameters? This is of particular interest, since the package only provides annotation for GRCh37.

Response: Thank you for your comment. We apologize for not making these details explicit. We have added more details on how these gene sets were collected and pruned in the Materials and Methods (currently lines 698-728 on page 23-24). Briefly, we collected genes sets from seven existing gene set databases. We then pruned the gene sets by filtering gene sets with small gene set size (≥ 20 genes for human gene sets, and ≥ 10 genes for gene signatures due to its small size). The pruning step of removing small sized gene sets is for computational reasons: many GSE algorithms including ours become unstable on gene sets with a small number of genes. Following your comment, we also mentioned this important reason for pruning in the Materials and Methods section (lines 728-731 on page 24), which was previously described with simulations in the Results section (currently lines 213-218 on page 8) and the Discussion section (currently lines 598-615 on page 19).

4. In GO, one gene can be a member of multiple groups, which complicates the testing due to non-independence. How does iDEA deal with this problem? Some examples of tests for nested groups would be interesting to see.

Response: Thank you for your comment. Previously we followed most existing GSE approaches and accounted for test non-independence due to gene set overlap through permutation. Specifically, we construct an empirical null distribution by permuting gene labels. Such permutation retains gene set overlap under the empirical null: if the gene set A contains genes that are overlapped with genes in the gene set B in the real data, then gene set A would also contain the same number of overlapped genes with the gene set B in the permuted data. Consequently, the test statistics on gene set A will be correlated with that on the gene set B in a similar fashion in the permuted data as in the real data. By estimating false discovery rate (FDR) based on such permuted null, we can account for test non-independence due to gene set overlaps. Certainly, other approaches to deal with GSE test when non-independence exist, and some of them are more *ad hoc* than others. For example, PADOG [3] introduces the idea of down-weighting genes that occur in a high frequency across the gene sets. However, simply down-weighting genes that

occur in many gene sets may not be ideal, as this approach may down-weight important genes that are key factors in the analyzed data set. Another method proposed by Jiang and Gentleman [4] examines pairs of gene sets one at a time. For each pair of gene set, Jiang's method divides genes into three categories: one category of genes that are only in the first gene set, one category of genes that are only in the second gene set, and one category of genes that are common in both gene sets. Afterwards, Jiang's method calculates three p-values, one for each category of genes. By computing p-values in each set, Jiang's method can explicitly deconvolute the results in the presence of gene set overlap. Therefore, Jiang's method can be potentially applied to the gene sets identified by iDEA to further dissect particular set of genes drive the enrichment signal. We have added discussion on the important issue of gene set overlap in line of above in the Discussion section (lines 578-579 on page 18, lines 587-597 on page 19).

In addition, we also performed new analysis to further examine the issue of gene set overlap in the real data applications. *First*, we found that gene set overlap appears to be moderate in the real data sets. For example, in the Human Embryonic scRNA-seq data, the pairwise overlapping between all gene sets we tested and the median overlapping genes between pairwise significant gene sets is 1, which is relatively small comparing to the median gene set size of 143. In the mouse sensory neuron scRNA-seq data, the median overlapping genes between pairwise significant gene sets is 5, which is also relatively small comparing to the median gene set size of 131. *Second*, we applied Jiang's method to analyze the top 50 gene sets identified by iDEA in order to further dissect particular set of genes that drive the enrichment signal. (Note that we did not apply to all significant gene sets due to the heavy computational burden of Jiang's method). In the human embryonic data, 692 of the 1,225 gene set pairs have higher than 20 genes in overlap. For each of these 692 gene set pairs in turn, we calculated the three p-values as mentioned in the previous paragraph. Among the total 2,076 adjusted p -values (Bonferroni correction) we calculated, 1,397 of them are less than 0.05. We first look at the intersection part, 35 out of 692 intersection sets have adjusted p -value is less than 0.05. For the disjoint parts, 1,362 out of 1,384 are significant. This observation suggests that among the top 50 significant gene sets we identified, gene set specific genes are significantly enriched, suggesting that it is not the overlapped genes that drive the enrichment signal and that gene set overlap does not appear to introduce excessive false signals. We further looked at the combination of the top first gene set GO:0001944 (vasculature development) (Table S10). From the table, we observed that the significance of this gene set is induced by both the overlapping parts and non-overlapping parts. Following the same procedure, we also applied Jiang's method to analyze the top 50 gene sets identified by iDEA in the mouse sensory neuron scRNA-seq data. 1,025 out of

1,225 gene set pairs have higher than 20 genes in overlap. For each of these 1,025 gene set pairs in turn, we calculated the three p-values as mentioned in the previous paragraph. Among the total 3,075 adjusted p -values (Bonferroni correction) we calculated, 2,603 of them are less than 0.05. We first look at the intersection part, 889 out of 1,025 intersection sets have adjusted p -value is less than 0.05. For the disjoint parts, 1,714 out of 2,050 are significant. We further looked at the combination of the top first gene set GO:0044425 (obsolete membrane part) (Table S11). From the table, we observed that the significance of this gene set is induced by both the overlapping parts and non-overlapping parts. These new results are briefly mentioned in the Discussion section (lines 579-587 on page 18-19), with details provided in section 6 in the Supplementary Text.

5. In the same vein, it is unclear how the authors construct gene sets based on 10xGenomics signatures. To my knowledge these signatures are mainly used for the classification of cells and it is unclear to me how these signatures would be used to infer the biological function of genes.

Response: Thanks for your comment. We apologize for not explaining these details in the previous version. For the 10xGenomics dataset analysis, we intentionally focused only on gene sets that were mainly used for cell type classification. These cell type specific gene sets genes were obtained from six other projects using the xCell package [5]. We chose to use this type of gene sets in the 10xGenomics data because these gene sets contain CD4+ and CD8+ cell type signatures and thus can be treated as true positives/golden standard for method comparison in real data sets, where we typically do not know the underlying truth. Therefore, these cell type specific gene sets, when further paired with the differential expression analysis between CD4+ and CD8+ cell types in the 10xGenomics data, provides us with a unique opportunity to validate the performance of our method and demonstrate its high power in the real data application. Certainly, while this approach has the benefits of knowing the true positive gene sets, it also has the drawback of not being able to infer the biological function of identified differentially expressed genes. Therefore, we demonstrated the biological function of identified differentially expressed genes in the other three data sets. We have now explicitly mentioned the rationale of using cell type specific gene sets in the 10xGenomics data (lines 401-403 on page 13, lines 441-446 on page 14).

In addition, we have also performed new analysis by using the large set of human gene sets. Briefly, we found that iDEA produces calibrated p -values under the permuted null and remains reasonably powerful in detecting enriched gene sets. While the percentage of significantly enriched gene sets (= 5.9%) in the large human gene sets base is smaller than that (= 17.4%) in the 10x Genomics signatures, the top enriched gene sets from the

former do contain important biological functions including metabolism of proteins [6] which is the pathway related to metabolic programs which support the differentiation of CD4 T helper cells and GO:0006605 (protein targeting) which is related to the biological mechanisms in the immune response and the protein targets of human CD4+ and CD8+ T cell distinct [7]. These new results are shown in Figure S27 and Table S8, with details provided in the Results section (lines 446-456 on page 14-15).

Minor comments:

1. In several figures where axes are labelled as p-value, but it is $-\log_{10}(\text{p-value})$?

Response: Thank you for spotting the error. The p-value in the qqplot is indeed under $-\log_{10}$ scale. We have modified the legends in the corresponding plots (Figure 2, Figure 4B, Figure 5B, Figure 6B, Figure S2, Figure S10 and Figure S16).

2. Supplementary Figure 13: Because iDEA increases the numbers the total number of detected genes, I find it difficult to compare the numbers in the Venn Diagrams. Could the authors provide some percentages or use upsetR plots?

Response: Thank you for your comment. Follow your suggestions, we have added new plots using upsetR [8] along with the previous Venn Diagrams. Please refer to the new supplementary Figure S14, Figure S20 and Figure S24.

Response to Reviewer #2

In this manuscript, the authors describe a new computational method for gene set enrichment analysis. The proposed iDEA algorithm uses a hierarchical Bayesian model with EM-MCMC estimation of the parameters to obtain p-values for enrichment of DE genes within each gene set. The authors go on to demonstrate improved performance compared to several other DE and GSEA methods on real and simulated data. I found the hybrid of frequentist and Bayesian concepts rather bizarre, but that notwithstanding, I found many other causes for concern with the statistical aspects of this work. These issues are described in more detail below.

Response: Thank you for your constructive comments. Our detailed responses are listed below.

MAJOR:

1. It seems that the authors are using their proposed method to detect differences between cell types. However, the effect sizes are often overestimated and the standard errors understated for such comparisons, as the "cell types" are not known *a priori* but are instead defined from the expression values. (A more detailed description of this effect is available at [http://bioconductor.org/packages/3.9/workflows/vignettes/simpleSingleCell/inst/doc/de.html#5_caveats_with_interpreting_de_\(p\)-values](http://bioconductor.org/packages/3.9/workflows/vignettes/simpleSingleCell/inst/doc/de.html#5_caveats_with_interpreting_de_(p)-values).) Thus, the authors' focus on error control in simulations is largely irrelevant to practical analyses, as the underlying effect sizes and their standard errors cannot be considered as unbiased estimates.

Response: Thank you for your comment. Indeed, in the case when the "cell types" are not known *a priori* and inferred from the whole gene expression matrix, then the DE analysis results might contain false signals with an enrichment of small DE p-values under the null. This is a common problem for all DE methods in scRNA-seq studies [9]. As far as we are aware of, no good solution is currently available to address this important issue for DE methods. Therefore, we followed the mainstream scRNA-seq DE papers and *did not* focus on the error control in DE analysis of these existing DE methods. Instead, we only focused on the error control of GSE analysis.

In addition, one of our real data contains cell types that are known *a priori* and not inferred from the whole expression matrix, while the other two data contain cell types that are extensively validated through approaches other than inferring based on the whole

expression matrix. Specifically, for the human embryonic stem cell scRNA-seq dataset, the cell types are obtained from fluorescence-activated cell sorting (FACS) analysis before mixing for scRNA-seq. FACS relies on known cell type markers and represents a somewhat unbiased strategy for cell type clustering [10]. For the mouse neuronal scRNA-seq dataset, the cell types are initially inferred through an iterative PCA-based procedure and are further validated by comparing the hierarchical relationship of the neuronal types with the known developmental origin of sensory neuron types, as well as by comparing neurons with distinct and characteristic soma sizes in their identified neuronal class. In addition, the inferred neuronal cell types are further confirmed by double and triple immunohistochemical staining (e.g. NP1 cell type by staining of PLXNC1). For the 10x Genomics PBMC scRNA-seq dataset, the identity of cell types was inferred by aligning cluster-specific genes to known markers of distinct PBMC populations as well as comparing against the transcriptomes of the purified populations in PBMC subsets. Their approach has been found to be largely consistent with conventional marker-based methods and the major cell types reach to the expected ratios in PBMCs. We have also displayed t-SNE plot in Figure S22, which clearly shows distinct cell clusters. Because the cell types in these data are either truly known or extensively validated through various other approaches, the DE analysis results are less likely influenced by the cell type inference step, at least as compared to other data that are fully relying on the whole gene expression matrix for cell type inference. Following your comments, we have added this important caveat for all DE methods in scRNA-seq studies when DE comparison is performed between cell types and when these cell types are also inferred based on the whole gene expression matrix in the Discussion section (lines 615-625 on pages 19-20).

2. The authors use the standard error of $\hat{\beta}$ in the variance term for their distribution of β in Equation 1. This makes little sense. The assumed distribution of the true effect size should not depend on the properties of its estimate; this is akin to saying that the truth changes depending on how we estimate it, which is generally absurd (quantum physics aside). From a practical perspective, the assumed distribution shrinks as the number of cells increases due to improved precision of the effect size. This seems as if it would reduce the power to identify DE genes, as the likelihood of a large effect size being generated from the normal component goes to zero. One would have instead hoped that more cells would provide greater power to detect DE.

Response: Thank you for your comment. We apologize for not explaining the model clearly which may have caused a potential miscommunication. The use of $se(\hat{\beta}_j)$ on modeling β_j in iDEA is equivalent to the following model based on the marginal z scores:

$$z_j \sim \pi_j N(0, \sigma_\beta^2 + 1) + (1 - \pi_j)N(0,1)$$

$$\text{logit}(\pi_j) = \log\left(\frac{\pi_j}{1-\pi_j}\right) = \tau_0 + a_j\tau_1.$$

where z_j is the marginal z-score on the DE evidence for the j -th gene. The above model practically assumes that, with proportion $1 - \pi_j$, j -th gene is not a DE gene and its z-score follows the null $N(0,1)$ distribution; with proportion π_j , j -th gene is a DE gene and its z-score follows an alternative $N(0, \sigma_\beta^2 + 1)$ with a variance greater than one. Our modeling of z-scores follows that of Efron et al [11] and Efron and Tibshirani [12] (and many others), where a mixed model was used for modeling these marginal z-scores. Certainly, as the reviewer correctly pointed out, modeling z-scores in all these models does have the caveat that the prior distribution of true effect sizes are dependent on the standard errors, and subsequently the sample size. Such caveat appears to be mild, as these z-score based models have been widely adopted and applied to a range of data sets with a variety of sample sizes. In our particular study, we also found that iDEA provided well controlled type I error with high power across all three real datasets, with sample sizes ranging from 243 to 1,458. In addition, such “caveat” of the prior dependence on sample size has been recently shown to have attractive theoretical properties (e.g. [13] and many follow up studies) – therefore, it is far from clear at the moment whether such prior dependence on sample size should be considered as a caveat or a desirable feature. Nevertheless, we have added a few sentences to point out the relationship to z-score modeling in the Materials and Methods section (lines 641-644 on page 21). We have also added a few sentences to point out this general caveat/feature of these z-score based models in the Discussion section (lines 539-552 on pages 17-18).

In addition, follow your suggestion, we have also developed a new model that does not have the aforementioned caveat/feature. The new model assumes that

$$\hat{\beta}_j = \beta_j + \epsilon_j, \epsilon_j \sim N(0, \text{se}(\hat{\beta}_j)^2)$$

$$\beta_j \sim \pi_j N(0, \sigma_\beta^2) + (1 - \pi_j)\delta_0$$

$$\text{logit}(\pi_j) = \log\left(\frac{\pi_j}{1-\pi_j}\right) = \tau_0 + a_j\tau_1.$$

where the prior of the true effect size β_j is no longer depend on the standard error $\text{se}(\hat{\beta}_j)$. We have implemented this iDEA variant into the IDEA software package. We have applied the new model to analyze the scRNA-seq data, where the new results largely consistent with that obtained from the z-score based model. The detailed model is provided in the Supplementary Text (section 4.1, pages 43-46). The new results using

this model are provided in the Supplementary Figures S31-34, with details available in the Discussion section (lines 553-567 on page 18).

3. The prior distribution for σ_{β} is poorly justified, and I could not find an analysis of the sensitivity of the method to changes in prior parameters. This is important as σ_{β} effectively determines the likelihood by which a gene is considered to be DE, which in turn serves as the basis of the gene set enrichment aspect of the method. In addition, the prior distribution for the indicator variables γ are not described, despite the fact that the authors are computing posterior distributions for $\gamma \in \{0, 1\}$ via MCMC.

Response: Thank you for your comments. For your first comment, we are a bit puzzled on why the reviewer believed the prior distribution for σ_{β} was poorly justified, given that an inverse gamma distribution is the conjugate distribution for a variance parameter and is thus commonly used in the literature. However, we do fully agree with the reviewer that the particular choices of the hyper-parameters (i.e. a_{β}, b_{β}) in the inverse gamma distribution worth more careful exploration. We did not explore different choices of hyper-parameters before, because it seems to us intuitively that the hyper-parameters would not matter much as long as there are a sufficient number of DE genes. Specifically, the inverse Gamma distribution serves as the prior for the variance parameter. The effect sizes across many DE genes likely contain majority of the information for estimating the variance variant and such information likely overwhelm the prior information contained in the inverse gamma distribution. To validate such intuition and examine the sensitivity of results with respect to the hyperparameters, we varied the hyperparameters and test across a range of gene sets with different coverages in our three real datasets. Specifically, we varied the hyperparameters from prior mean of gamma to be 0.001, 0.1, 1, 10, 100 and varied the coverage rate to be the 10th, 30th, 50th, 70th, 90th percentile of the gene set size of the gene sets we used for corresponding real data analysis in our manuscript. For example, for the human embryonic scRNA-seq dataset, we pick the gene set with coverage rate to be the 10th, 30th, 50th, 70th, 90th percentile of the gene set size of the human gene sets we analyzed and set the hyper parameter in the prior distribution of $\sigma_{\beta}^2, (a_{\beta}, b_{\beta})$ to be (3, 0.02), (3, 0.2), (3, 2), (3, 20), (3, 200) respectively. Thus, creating a different set of prior distributions with a wide range of mean. For all the three real datasets, the estimate of both gene set enrichment coefficient and its variance do not vary too much across a wide range of prior distributions of σ_{β}^2 . Therefore, it does seem that results are all reasonably insensitive to the choice of these hyperparameters. The new results are shown in the Supplementary Figure S28, with details explained in the Materials and Methods section (lines 929-950

on pages 29-30) and the Results section (lines 457-462 on page 15). We also added a short sentence to explain that we chose an inverse gamma distribution due to conjugacy (lines 648-649 on page 21).

For your second comment, we apologize for not making the prior distribution for the indicator variables γ_j clearer. We previously only mentioned that the vector of binary indicators $\boldsymbol{\gamma} = (\gamma_1, \dots, \gamma_p)^T$ indicate whether each gene is a DE gene ($\gamma_j = 1$) or not ($\gamma_j = 0$) (currently lines 659-660 on page 22) and that the gene-specific probability of being a DE gene as π_j ; we previously thought these information were sufficient for describing the prior distribution γ_j , given that such indicator variables are commonly used for variable selection models in the literature. Following your comment, we now explicitly mention that γ_j follows a Bernoulli distribution (e.g. as in [14])

$$\gamma_j \sim \text{Bern}(\pi_j)$$

We have added a sentence on this important information in the Materials and Methods section (lines 661-662 on page 22).

4. It is not clear to me how the proposed method considers a gene to be DE if the calculations are always performed in the context of gene sets. Are the authors calling genes to be DE if they are called as DE in any calculation with any gene set? This seems to run into major problems with multiple testing.

Response: Thank you for your comment. We apologize for not explaining this clearly in the previous manuscript. As we explained in the previous manuscript, DE analysis in the real data is performed based on a pre-selected gene set. The pre-selected gene set is based on biological knowledge, and such prior knowledge would provide valuable prior information for DE analysis and potentially improve the reliability, consistency and power of DE analysis. Thus, for DE analysis in the real data sets, we mainly examined the results using a pre-selected gene set with known relevance to the data. Specifically, for the human embryonic stem cell scRNA-seq dataset, we included the gene set *vasculature development* which is known to be important for vasculature progression and endothelial cell development. For the mouse sensory neuron scRNA-Seq dataset, we included the gene set GO:0097458 (neuron part) which is known to be relevant to the nervous system. For the 10x Genomics PBMC scRNA-seq dataset, we included the gene set CD8+ T-effector memory, which known to be relevant to CD8 T cells functions. These information were available in the Results section (currently lines 317-318 on page 11, lines 378-379 on page 13, lines 418-419 on page 14) and Methods and Materials section (currently lines 876-878 on page 28, lines 899-900 on page 29, lines 913-915 on page 29).

Certainly, the above strategy has the benefits of allowing the biologists to select important gene sets that are known to be important in the experiment, thus taking advantage of the decades of prior molecular biology research. However, it is challenging to carry out the above strategy if the biologists do not have prior information on which gene set is important. For the latter case, we have now developed a new strategy to aggregate DE evidence on a particular gene across all gene sets through Bayesian model averaging. Specifically, for the given gene, we denote its posterior inclusion probability (PIP) obtained using the gene set k as PIP_k . The corresponding Bayes factor quantifying its DE evidence based on the gene set k is $BF_k = PIP_k / (1 - PIP_k)$. With equal prior weights on different gene sets, the average Bayes factor quantifying its DE evidence based on all K gene sets is thus $ABF = \frac{1}{K} \sum_{k=1}^K BF_k$, which can be converted back to a posterior inclusion probability as $PIP = ABF / (1 + ABF)$. We found that PIPs computed this way is highly correlated with the PIPs computed based on the pre-selected gene set for majority of genes (Figure S35). We now provide both options for computing PIPs for quantifying DE evidence: biologists can choose to use pre-selected gene sets that are known to be relevant to the particular experiments, as is the case for all the real data applications; alternatively, biologists also have the option of using the Bayesian model averaging when such prior knowledge is not available. The new results are shown in the Supplementary Figure S35, with details explained in the Supplementary Text (section 5 on page 46) and the Discussion section (lines 568 – 577 on page 18).

5. The authors criticize existing methods as treating DE and GSEA analyses as two separate steps. However, it seems that the same thing is done here! The proposed method relies on the effect size and the standard error, which can be trivially transformed into a per-gene p-value. At this point, the DE analysis has effectively been performed already, so iDEA is really just doing the second-stage gene set enrichment analysis.

Response: Thank you for your comment. We apologize for not explaining our method well, which caused this unfortunate miscommunication. Unlike what the reviewer perceived, the DE analysis of iDEA is *not* performed based on the per-gene p-value calculated using the effect size estimate and its standard error. Instead, iDEA effectively performs an interactive procedure that iterates between the GSE analysis (based on testing τ_1) and the DE analysis (based on $P(\gamma_j = 1 | data)$). Different from traditional GSE methods which treat GSE analysis as a separate analytic step after DE analysis, iDEA integrates these two analyses together since GSE analysis and DE analysis are interconnected with each other statistically. iDEA is not just doing the second-stage gene set enrichment analysis here. Instead, whether the gene set is enriched or not in the GSE

analysis can provide valuable information for DE analysis in iDEA, and the more accurate DE information could serve as a better feedback in the GSE analysis to improve the power and ensure results reliability. Therefore, different from previous methods, iDEA updates DE results in addition to doing GSE analysis. We previously explained these important technical details in the method overview section in the Materials and Methods, with further details provided in the Supplementary Text. Following your comment, we have re-read the text but couldn't figure out which part might have led to reviewer's misunderstanding. This comment also somewhat contradicts with your valuable comment #4, which seems to understand that the DE analysis in iDEA is based on information from the GSE analysis. We would really appreciate if you could please let us know which parts of our manuscript lead to this important miscommunication, so that we can modify the text accordingly.

6. In the proposed EM-MCMC step, I do not understand how the posterior probabilities can be used as the outcome of a logistic regression. Surely the response must be categorical - is the logistic model's likelihood even valid for continuous values? If it is valid, that leads to the next question - are the standard errors of the posterior probabilities taken into consideration by the subsequent inference steps, or are users expected to run the MCMC for sufficient iterations that the standard errors are assumed to be close to zero?

Response: Thank you for your comment. We apologize for not explaining these technical details clearly in the previous manuscript. The logistic model's likelihood is indeed valid for continuous values (please refer to the log likelihood function in equation S1 in section EM-MCMC Inference Algorithm in the Supplementary Text; the logistic model is part of this log likelihood function). In addition, there is no need to take into account the standard errors of the posterior probabilities, since the expectation step in the EM algorithm only requires the expected values but not the standard errors (equation S2 in section EM-MCMC Inference Algorithm in the Supplementary Text). Following your comment, we have modified the corresponding sentences in the Materials and Methods section to make these points clear (lines 671-680 on page 22).

7. The authors focus heavily on the number of significant gene sets in their analyses of real datasets, which serves as evidence for iDEA's improved power over existing methods. However, as per point 1, this is largely irrelevant. The main focus is instead on the ranking of the top gene sets, as only these are used for biological interpretation. Even if we detect >1000 significant gene sets, no analyst will look at the 1000th gene set if the first ~100 sets are not useful for determining cell type or state.

Response: Thank you for your comment. We apologize for not explaining well the rationale for examining statistical power in the previous manuscript, which caused this unfortunate miscommunication. The number of significant gene sets at the given false discovery rate is perhaps the most objective criterion for evaluating and comparing statistical power of different methods in real data. This criterion is commonly used in almost all existing statistical literature and serves as the golden standard for unbiased and subjective evaluation of different methods in terms of statistical power. Therefore, we follow the mainstream of statistical literature and focus heavily on the number of significant gene sets for power comparison in the real datasets. In addition, your point #1 is related to marginal DE analysis and it is unclear how this point is relevant to the discussion of GSE analysis here. Certainly, we agree with the reviewer that there are currently only resources available to evaluate the first ~100 sets in the case when there are >1000 significant gene sets. Therefore, we previously examined the top ~100 gene sets and illustrated many of them in the main text. For example, for each real dataset, we checked the biological significance of identified gene sets by searching the related literatures. We have shown that our top ~100 identified gene sets are highly correlated with the compared cell type functions and characteristics (currently lines 302-314 on pages 10-11, lines 362-377 on pages 12-13 and 410-415 on pages 13-14).

In addition, we hope the reviewer would also agree with us that the statistical power of different methods effectively determines how much true positives and false positives one would get in this ~100 set. More importantly, statistical power would determine how many significant gene sets we would get, if any, in the challenging cases where it is extremely hard to identify enriched gene sets (e.g. comparing similar cell types CD4+ and CD8+ in 10x Genomics PBMC scRNA-seq dataset). Therefore, we focus on statistical power in the real datasets by examining the number of significant gene sets. Following your comment, we have explained the rationale and importance of examining statistical power in the real data in the Results section (lines 294-295 on page 10).

8. The authors criticize methods like FGSEA and CAMERA on the basis of their inability to detect differences between genes inside and outside the test set. However, it seems that related criticisms would apply to iDEA. For example, iDEA would not detect a gene set where the enrichment of DE genes is the same in terms of the number of genes, but the effect sizes of the genes in the set are systematically higher than those outside of the set. One might say that detecting such differences is not the purpose of iDEA, but in that case, it would be more appropriate to compare iDEA to conceptually similar tests like the hypergeometric test and its derivatives (e.g., goseq).

Response: Thank you for your comment. Follow your suggestion, we have performed new analysis and compared iDEA with the hypergeometric test in GSE analysis for all three real scRNA-seq datasets. Since goseq package only provide gene sets from Gene Ontology Terms and KEGG pathway and does not provide user defined pathways to test, we constructed hypergeometric test directly on the same gene sets we analyzed for fair comparison. The new results are shown in the Supplementary Figure S13, Figure S19 and Figure S23 and are largely consistent with the main results. Specifically, across all three real datasets, iDEA performs better in terms of type I error control and is more powerful in identifying enriched gene sets compared with hypergeometric test. For example, hypergeometric test can identify only 1 gene set in the 10x Genomics PBMC dataset while iDEA identified 25 gene sets at a fixed FDR of 5%. The new results are shown in the Supplementary Figure S13, Figure S19 and Figure S23 with details provided in the Materials and Methods section (lines 762-766 on page 25) and the Results section (lines 300-302 on page 10, lines 361-362 on page 12, and lines 415-417 on page 14).

9. I could not obtain p-values from the software when testing on a small simulated example. Running code adapted from <https://xzhoulab.github.io/iDEA/> just gave me an empty output data.frame in idea@gsea.

```
> library(iDEA)
>
> set.seed(100)
> ngenes <- 10000
> summary_data <- cbind(
> log2FoldChange=rnorm(ngenes),
> lfcSE2=rchisq(ngenes, 1)/100
> )
> rownames(summary_data) <- sprintf("GENE_%i", seq_len(ngenes))
>
> annotation_data <- matrix(0, ngenes, 2)
> annotation_data[head(order(summary_data[,1]), 100),1] <- 1
> annotation_data[sample(ngenes, 100), 2] <- 2
>
> colnames(annotation_data) <- LETTERS[1:ncol(annotation_data)]
```

```
> rownames(annotation_data) <- sprintf("GENE_%i", seq_len(ngenes))  
>  
> idea <- CreateiDEAObject(summary_data, annotation_data, num_core=2)  
> idea <- iDEA.fit(idea)  
> idea <- iDEA.louis(idea)
```

Response: Thank you and sorry for the error. We have corrected the error and updated both the package and tutorial which should work and generate the outputs of `idea@gsea`.

MINOR:

- **The use of the `expit()` transformation in the simulation design is not explained. Why not use a simpler approach where a given proportion of genes are set as DE?**

Response: Thank you. Following your first comment, we have now added more detailed explanation on the `expit()` function in the methods overview and simulation design subsection (lines 155-157 on pages 6-7). For your second comment, we did not set a given proportion of genes to be DE because such proportion will likely change across data sets [15]. Therefore, we rely on the iDEA modeling framework to infer such proportion based on the data at hand.

Response to Reviewer #3

Sun et al. propose iDEA, a unified statistical framework that jointly performs differential expression (DE) and gene set enrichment (GSE) analysis. The method is certainly welcome in the field of transcriptomic studies. Analysis is carried out to evaluate performance via both simulations and empirical studies. The paper is overall well written. My comments are below.

Response: Thanks for your positive review and constructive comments. Our detailed responses are provided below.

1. The summary statistics from DE analysis are used as input for iDEA. I wonder why the authors restrict their analysis/method to the single-cell setting. Of course, the model for scRNA-seq would be much different than that for bulk RNA-seq, as the authors have pointed out. However, the noise modeling etc specific to scRNA-seq is handled by the respective existing methods adopted (e.g., MAST, zingeR). The distinguishing features of iDEA that specific to scRNA-seq need to be clearly stated and emphasized.

Response: Thank you for your comment. Indeed, the modeling framework of iDEA is flexible and is not restricted to scRNA-seq studies. We focused mainly on scRNA-seq studies, because, in addition to developing a new method, we also aim to perform the first comprehensive comparative study on GSE methods for scRNA-seq studies. Our study revealed some important features of GSE analysis on scRNA-seq. For example, we found that gene correlation does not appear to influence GSE results much in the scRNA-seq data as compared to bulk RNA-seq data, presumably because the technical difficulties such as dropout events made scRNA-seq data sparser and noisier, rendering gene-gene correlation harder to estimate well. As another example, the large scale of scRNA-seq datasets made some of the GSE methods (e.g. the standard GSEA) suffer from heavy computational burden. In addition, use of summary statistics makes iDEA especially suitable for scRNA-seq studies where different statistical models are needed for modeling data obtained from different scRNA-seq techniques. It is well known that different individual-level data modeling assumptions work preferentially well for different types of scRNA-seq data sets collected from various sequencing techniques and diverse experimental designs. As a result, using summary statistics allows iDEA to be applicable to a wide range of scRNA-seq data types currently modellable with existing DE methods. Following your comment, we have re-organized and merged several relevant paragraphs in the previous Discussion section into a comprehensive paragraph there to make it more clear the rationale behind the focus on scRNA-seq studies (lines 484-521 on pages 16-17).

We also fully agree with the reviewer that the method iDEA itself indeed can be equally applied to bulk RNA-seq studies. Following your comment, we have performed additional analysis and added an application example on bulk RNA-seq. The results from bulk RNA-seq are largely consistent with scRNA-seq applications, demonstrating the calibrated type I error control and higher power of iDEA. The bulk RNA-seq results are shown in the new Supplementary Figure S30, with detailed provided in the Discussion section (lines 521-526 on page 17).

2. Across all simulation runs and real data analyses, iDEA returns more significant genes/gene sets. The authors only looked at type I error control via QQ plots of p-values and showed that the p-values by iDEA were well calibrated. (BTW, Lambda_gc, the genomic control factor, needs to be explicitly defined.) What are the actual type I error rates, since this is from simulation run? Also, it has been well acknowledged that model-based simulated sc data do not reflect the true observed sc data. The authors need to demonstrate iDEA controls for type I error by performing test under the true null using two data sets of the same cell types/tissues/model organisms.

Response: Thank you for your comments. Following *your first suggestion*, we have added the detailed definition of lambda_gc (lines 191-194 on pages 7-8 in the Results section; lines 848-854 on page 27 in the Materials and Methods section). We are a bit puzzled by *your second question* “What are the actual type I error rates, since this is from simulation run?”. In the simulations, since we know the truth, we plotted the p-values (on -log₁₀ scale) against the actual type I error rates (on -log₁₀ scale) in all QQ plots under the null. These QQ plots were used to explore a wide range of type I error rates (from 1e-4 to 1.0) instead of focusing on a particular type I error rate. The actual type I error rates for all p-values are displayed on the x-axis in these QQ plots. We felt using QQ plots are much easier to visualize type I error control across a range of type I error rates and contain much more information than a type I error table, but we would be more than happy to also include standard type I tables if you felt that could be useful.

Finally, we previously only examined type I error control of different GSE methods in the real datasets through permutations. Following *your third suggestion*, we have performed new analyses to further examine type I error control of different GSE methods in the real data through comparing tests on the same cell types. To do so, in each of the three real data sets, we split the cells within the same cell type into two sets with an equal number of cells. The cell types we examined include the DEC cell type in the human embryonic stem cell data; the NP1 cell type in the mouse sensory neuron data; and the CD8+ cell type in the 10x Genomics data. Note that this approach leads to a null distribution of DE

p-values, which is a stronger condition than our previous permutations which only aimed to create a null distribution of GSE p-values. We then examined type I error rates of different GSE methods in these at different p-value cutoffs. Consistent with null permutations in the real data, iDEA controls type I error well in the first data and become conservative in the second and third datasets. PAGE generally displays inflation in type I error control across all three real datasets while GSEA and fGSEA display inflation especially when the p-value cutoff is larger. In contrast, CAMERA shows severe deflation in the first and second real dataset while inflation in the third dataset. These new results are shown in the new Supplementary Figure S29, with details provided in the Results section (lines 462-473 on page 15).

3. The two human empirical studies adopted drastically different number of gene sets (12,033 vs 144). Based on this, I have a few comments/concerns.

a. Do the users need to prune the gene sets themselves? Guidelines and discussions on this will be helpful.

Response: Thank you for your comment. We apologize not making the rationale of using different gene sets in the two data sets explicit. For the human embryonic stem cell data, we performed standard GSE analysis using traditional gene sets (hence the large number). Our goal was to use this data to illustrate how iDEA and other GSE methods perform in standard GSE analysis for scRNA-seq data and illustrate how the standard GSE analysis may provide us with new biological insights. The identified top enriched gene sets all make biological sense. However, we unfortunately cannot be sure whether these identified enriched gene sets are all true positives, since we do not know the truth in this real data set. Therefore, we resorted to use a small set of specific gene sets in the 10xGenomics data in order to directly examine the power of different GSE in that real data. Specifically, the small set of gene sets we used in 10x Genomics data contain CD4+ and CD8+ cell type signatures, and these signatures can be treated as true positives/golden standard in this real data. These cell type specific gene sets, when further paired with the differential expression analysis between CD4+ and CD8+ cell types in the 10xGenomics data, provides us with a unique opportunity to validate the performance of our method and demonstrate its high power in the real data application that otherwise would be challenging to do. Therefore, we used two different sets of gene sets (12,033 vs 144) in these two real data applications to serve as two different purposes. We have now provided these rationale in the updated Results section (lines 401-403 on page 13, lines 441-443 on page 14). For your specific question, users do not need to collect gene sets or filter them by themselves for standard GSE analysis. The standard set of human gene sets from seven databases are provided in our software package.

Certainly, users are always welcome to provide their own gene sets to answer specific biological questions.

b. The 12,033 gene sets were compiled from seven different databases. This number is on the same order of magnitude as the number of genes. Are there overlaps between the gene sets? Will this bias the FDR control procedures, i.e., will the testing across different gene sets be non-independent?

Response: Thank you for your comment. Indeed, there are overlaps between gene sets, but the overlap appears to be generally small: the median number of overlapped genes among pairs of gene sets in the set of 12,033 gene sets is only 1, as compared to the median gene set size of 143. In addition, we have added new analysis to carefully examine the top identified enriched gene sets in the real data applications. The new results suggest that gene set overlap does not appear to introduce excessive false signals (Table S10-S11; details in section 6 in Supplementary Text). Nevertheless, we followed most existing GSE approaches and accounted for GSE test non-independence due to gene set overlap through permutation of gene labels. Such permutation retains the gene set overlap proportion under the empirical null: if one gene set contains genes that are overlapped with genes in another gene set in the real data, then the first gene set would also contain the same number of overlapped genes with the second gene set in the permuted data. Consequently, the test statistics on the two gene sets would be correlated in a similar fashion in the permuted data as in the real data. By estimating FDR based on such permuted null, we can account for test non-independence due to gene set overlaps. Certainly, other approaches to account for gene set overlap also exist [3] [4]. Following your comment, we have added these new results in Table S10-S11, with details provided in the Supplementary Text (section 6) and the Discussion section (lines 578-597 on pages 18-19).

c. For the 10X Genomics dataset, if using all 12,033 gene sets, will it be underpowered?

Response: Thank you for your comment. Following your comment, we have performed new analysis using the large set of 12,033 gene sets. As expected, iDEA produces calibrated p-values under permuted null and remains reasonably powerful in detecting enriched gene sets. Specifically, iDEA identified 421 significantly enriched gene sets among 12,033 at an FDR of 0.05. The top enriched gene sets identified by iDEA is also biologically meaningful. For example, metabolism of proteins [6] is the pathway related to metabolic programs which support the differentiation of CD4 T helper cells. GO:0006605 (protein targeting) is related to the biological mechanisms in the immune response and the protein targets of human CD4+ and CD8+ T cell distinct [7]. However, as expected

and as the reviewer also pointed out, the percentage of the significantly enriched gene sets (=5.9%) in the large gene sets is smaller than that (=17.4%) in the small gene sets – indeed, the 144 gene sets were selected specifically to validate the power of iDEA in this particular data. We have now added these new results in the Figure S27, with details provided in the Results section (lines 446-456 on pages 14-15).

d. For the 144 gene sets, the authors mentioned “we focused on examining a small set of 144 gene sets that contain important gene signatures of immune and stroma cell types...” Is there a cyclic problem? Usually, the gene sets are defined specific to both the tissue of interest and the study design (what are the treatment etc), and the gene set enrichments can be identified through the DE results. Here, the authors implicitly imposed a “prior” (by focusing on a small number of related gene sets based on, e.g., previous DE results), performed the analysis using this “prior”, and then tested the DE based on results. Will the results be simply passed down from the strong “prior”?

Response: Thank you for your comment. Indeed, while using 144 gene sets provide a unique opportunity to examine the GSE power of iDEA in the real data application, using these gene sets does run into cyclic problem and may cause the issue of an artificially higher DE power of iDEA in this real data. Following your comment, we have added a new DE analysis with iDEA but without adding any gene set information. We found that the number of DE genes identified by iDEA without the gene annotation (=252, at an FDR of 1%) is similar to the results obtained by adding the important gene set from these cell type defined gene sets (=255), both are larger than that identified by zingeR (=221). The new results are shown in Figure S26, with details provided in the Results section (lines 436-440 on page 14).

4. The GitHub page contains only functions with very limited and sketchy descriptions. For this method to be suited for publication at Nature Communications, a package needs to be compiled with detailed documentation and user manual, preferably with toy datasets.

Response: Thank you for your comment. We have updated the software GitHub page, with detailed documentation and user manual, together with a toy data set. Please refer to the details on <https://xzhoulab.github.io/iDEA/>.

5. On line 80, the authors discussed the limitations of existing univariate DE methods. It sounded as if iDEA would adopt a multivariate model, which is not true – iDEA still takes as input testing results based on univariate models.

Response: Thank you for your comment and we apologize for not making our points clear. We only intended to use that introduction section to say that iDEA can jointly model all genes together by borrowing information across genes in terms of DE effect size distribution properties. This is to contrast with the previous univariate approaches that analyze one gene at a time. We did not intend to say that iDEA would adopt multivariate regression type models that could also take into account the gene-gene correlations. We have modified that Introduction paragraph to make our points more clear (lines 71-74 on page 3).

6. Are the hyperparameters for the inverse Gamma fixed? Can they be empirically estimated and thus method/dataset specific?

Response: Thank you for your comment. Indeed, we fixed the hyperparameters for the inverse Gamma distribution because there is not enough information to estimate these parameters. Specifically, the inverse Gamma distribution serves as the prior for the variance parameter. While the effect sizes across many DE genes can be used to estimate the variance parameter, there is only one variance parameter, so it becomes impossible to estimate the hyperparameters for the inverse Gamma distribution. However, your comment does raise an important issue that we previously ignored; that is, whether the results from iDEA might be sensitive to the hyperparameters. To examine sensitivity of results with respect to the hyperparameters, we varied the hyperparameters and test across a range of gene sets with different coverages in our three real datasets. Specifically, we varied the hyperparameters from prior mean of gamma to be 0.001, 0.1, 1, 10, 100 and varied the coverage rate to be the 10th, 30th, 50th, 70th, 90th percentile of the gene set size of the gene sets we used for corresponding real data analysis in our manuscript. For example, for the human embryonic scRNA-seq dataset, we pick the gene set with coverage rate to be the 10th, 30th, 50th, 70th, 90th percentile of the gene set size of the human gene sets we analyzed and set the hyper parameter in the prior distribution of $\sigma_{\beta}^2, (a_{\beta}, b_{\beta})$ to be (3, 0.02), (3, 0.2), (3, 2), (3, 20), (3, 200) respectively. Thus, creating a different set of prior distributions with a wide range of mean. For all the three real datasets, the estimate of both gene set enrichment coefficient and its variance do not vary too much across a wide range of prior distributions of σ_{β}^2 . Therefore, it seems that results are all reasonably insensitive to the choice of these hyperparameters. This is perhaps not surprising given that the effect sizes across many DE genes likely contain majority of the information for estimating the variance variant and such information likely overwhelm the information contained in the inverse Gamma distribution. The new results are shown in the Supplementary Figure S28, with details explained in the Results section (lines 457 – 462 on page 15).

REFERENCES

1. Fan J, Salathia N, Liu R, Kaeser GE, Yung YC, Herman JL, Kaper F, Fan JB, Zhang K, Chun J, Kharchenko PV: **Characterizing transcriptional heterogeneity through pathway and gene set overdispersion analysis.** *Nat Methods* 2016, **13**:241-244.
2. Buettner F, Pratanwanich N, McCarthy DJ, Marioni JC, Stegle O: **f-scLVM: scalable and versatile factor analysis for single-cell RNA-seq.** *Genome Biol* 2017, **18**:212.
3. Tarca AL, Draghici S, Bhatti G, Romero R: **Down-weighting overlapping genes improves gene set analysis.** *BMC Bioinformatics* 2012, **13**:136.
4. Jiang Z, Gentleman R: **Extensions to gene set enrichment.** *Bioinformatics* 2007, **23**:306-313.
5. Aran D, Hu Z, Butte AJ: **xCell: digitally portraying the tissue cellular heterogeneity landscape.** *Genome Biol* 2017, **18**:220.
6. Buck MD, O'Sullivan D, Pearce EL: **T cell metabolism drives immunity.** *J Exp Med* 2015, **212**:1345-1360.
7. Rivino L, Kumaran EA, Jovanovic V, Nadua K, Teo EW, Pang SW, Teo GH, Gan VC, Lye DC, Leo YS, et al: **Differential targeting of viral components by CD4+ versus CD8+ T lymphocytes in dengue virus infection.** *J Virol* 2013, **87**:2693-2706.
8. Conway JR, Lex A, Gehlenborg N: **UpSetR: an R package for the visualization of intersecting sets and their properties.** *Bioinformatics* 2017, **33**:2938-2940.
9. Zhang JM, Kamath GM, Tse D: **Valid post-clustering differential analysis for single-cell RNA-Seq.** *Available at SSRN 3378005* 2019.
10. Baron CS, Barve A, Muraro MJ, van der Linden R, Dharmadhikari G, Lyubimova A, de Koning EJP, van Oudenaarden A: **Cell Type Purification by Single-Cell Transcriptome-Trained Sorting.** *Cell* 2019, **179**:527-542 e519.
11. Efron B: *Empirical Bayes analysis of a microarray experiment.* Stanford, Calif.: Division of Biostatistics, Stanford University; 2001.
12. Efron B, Tibshirani R: **Empirical bayes methods and false discovery rates for microarrays.** *Genet Epidemiol* 2002, **23**:70-86.
13. Narisetty NN, He X: **Bayesian variable selection with shrinking and diffusing priors.** *The Annals of Statistics* 2014, **42**:789-817.
14. George EI, McCulloch RE: **Variable selection via Gibbs sampling.** *Journal of the American Statistical Association* 1993, **88**:881-889.
15. Crow M, Lim N, Ballouz S, Pavlidis P, Gillis J: **Predictability of human differential gene expression.** *Proceedings of the National Academy of Sciences* 2019, **116**:6491-6500.

Reviewers' Comments:

Reviewer #1:

Remarks to the Author:

All of my concerns have been addressed.

Reviewer #2:

Remarks to the Author:

The authors have addressed all of my concerns. I am still a bit bemused about my points 2 and 6, but they are not a big deal. Some misgivings remain about the practical utility of increased power for gene set testing, but I will give the software some benefit of the doubt in this regard.

Reviewer #3:

Remarks to the Author:

The authors have addressed my previous concerns.

REVIEWERS' COMMENTS:

Reviewer #1

All of my concerns have been addressed.

Response: Thank you very much.

Reviewer #2

The authors have addressed all of my concerns. I am still a bit bemused about my points 2 and 6, but they are not a big deal. Some misgivings remain about the practical utility of increased power for gene set testing, but I will give the software some benefit of the doubt in this regard.

Response: Thank you very much.

Reviewer #3

The authors have addressed my previous concerns.

Response: Thank you very much.